# Perfect secrecy cryptography via mixing of chaotic waves in irreversible time-varying silicon chips

A. Di Falco [1,4], V. Mazzone[2,4], A. Cruz[3] & A. Fratalocchi[2*]

Protecting confidential data is a major worldwide challenge. Classical cryptography is fast and scalable, but is broken by quantum algorithms. Quantum cryptography is unclonable, but requires quantum installations that are more expensive, slower, and less scalable than classical optical networks. Here we show a perfect secrecy cryptography in classical optical channels. The system exploits correlated chaotic wavepackets, which are mixed in inexpensive and CMOS compatible silicon chips. The chips can generate 0.1 Tbit of different keys for every mm of length of the input channel, and require the transmission of an amount of data that can be as small as 1/1000 of the message's length. We discuss the security of this protocol for an attacker with unlimited technological power, and who can access the system copying any of its part, including the chips. The second law of thermodynamics and the exponential sensitivity of chaos unconditionally protect this scheme against any possible attack.

[1] School of Physics and Astronomy, University of St. Andrews, North Haugh, St. Andrews KY16 9SS, UK. [2] PRIMALIGHT, Faculty of Electrical Engineering, Applied Mathematics and Computational Science, King Abdullah University of Science and Technology, Thuwal 23955-6900, Saudi Arabia. [3] Center for Unconventional Processes of Sciences (CUP Science), 6475 E Pacific Coast Highway, Los Angeles, CA 90803, USA. [4]These authors contributed equally: A. Di Falco, V. Mazzone. *email: andrea.fratalocchi@kaust.edu.sa

With an information society that transfers an increasingly large amount of personal data over public channels, information security is an emerging worldwide challenge[1,2]. Conventional cryptographic schemes based on data encryption standard (DES), advanced encryption standard (AES), and Rivest, Shamir, and Adleman (RSA) encode messages with public and private keys of short length. The main advantage of these algorithms is speed, and the main disadvantage is their security, which relies on computational and provable security arguments and not on unconditional proofs. A major threat lies in the development of quantum computers, which are predicted to crack any of these ciphers in a short period of time[3].

A perfect secrecy cryptography, known as a one-time pad (OTP) was invented at the time of the telegraph and then patented by Vernam[4–6]. The Vernam cipher encodes the message via a bitwise XOR operation with a random key that is as long as the text to be transmitted, never reused in whole or in part, and kept secret. Shannon demonstrated that this scheme, properly implemented, is unbreakable and does not offer any information to an attacker, except the maximum length of the message[6]. Almost a century later, despite its proven absolute security, the OTP is still not adopted for lack of a practical and secure way for users to exchange the key.

Since the 1980s, research efforts have been dedicated towards solving this problem with point-to-point quantum key distribution (QKD) algorithms, which leverage on the unclonability of single photons[7]. While the progress of QKD in the past decades has been enormous[8–12], there are still critical challenges derived by the limits of quantum communications[13–19]. Due to the impossibility of amplifying single photons[20], quantum networks are currently unable to scale up globally; their data transfer is considerably slower than classical optical communications, which

already count with hundreds of high-bandwidth intercontinental lines, communication speed close to the light limit, and massive investments for the next years[21–26].

Here we develop a physical realization of the OTP that is compatible with the existing optical communication infrastructure and offers unconditional security in the key distribution.

## Results

**Protocol scheme of Vernam cipher on classical channels.** It is well known that chaos generates time varying signals that are mathematically unpredictable[27,28]. This originates from the sensitivity to input conditions: two nearby states $\mathbf{x}(t=0)$ and $\mathbf{x}'(0) = \mathbf{x}(0) + \epsilon$, even when $\epsilon \to 0$, always originate exponentially diverging trajectories $\Delta(t) = |\mathbf{x}(t) - \mathbf{x}'(t)| \sim e^{\mu t}$, with $\mu$ the largest Lyapunov coefficient[27]. By leveraging on this property, we show that it is possible to create a bidirectional communication channel for securely exchanging random keys of arbitrary length.

In this system (Fig. 1a), the two users—Alice and Bob—possess two chips that generate chaotic light states that are transmitted on a public classical optical channel. Each light state, indicated as $A_n$ for Alice and $B_{n'}$ for Bob, is a random superposition of optical waves[29] at different frequencies:

$$A_n = \sum_m a_{nm} \cos(\omega_m t + \phi_{nm}), \quad B_{n'} = \sum_m b_{n'm} \cos(\omega_m t + \psi_{n'm}),$$

(1)

with uncorrelated random amplitudes $a_{nm}$, $b_{n'm}$, and phases $\phi_{nm}$, $\psi_{n'm}$. These states are generated from the chaotic scattering of broadband pulses with different frequencies $\omega_1, ..., \omega_m$, and diverse input conditions $n$ and $n'$ (position, angle, polarization,

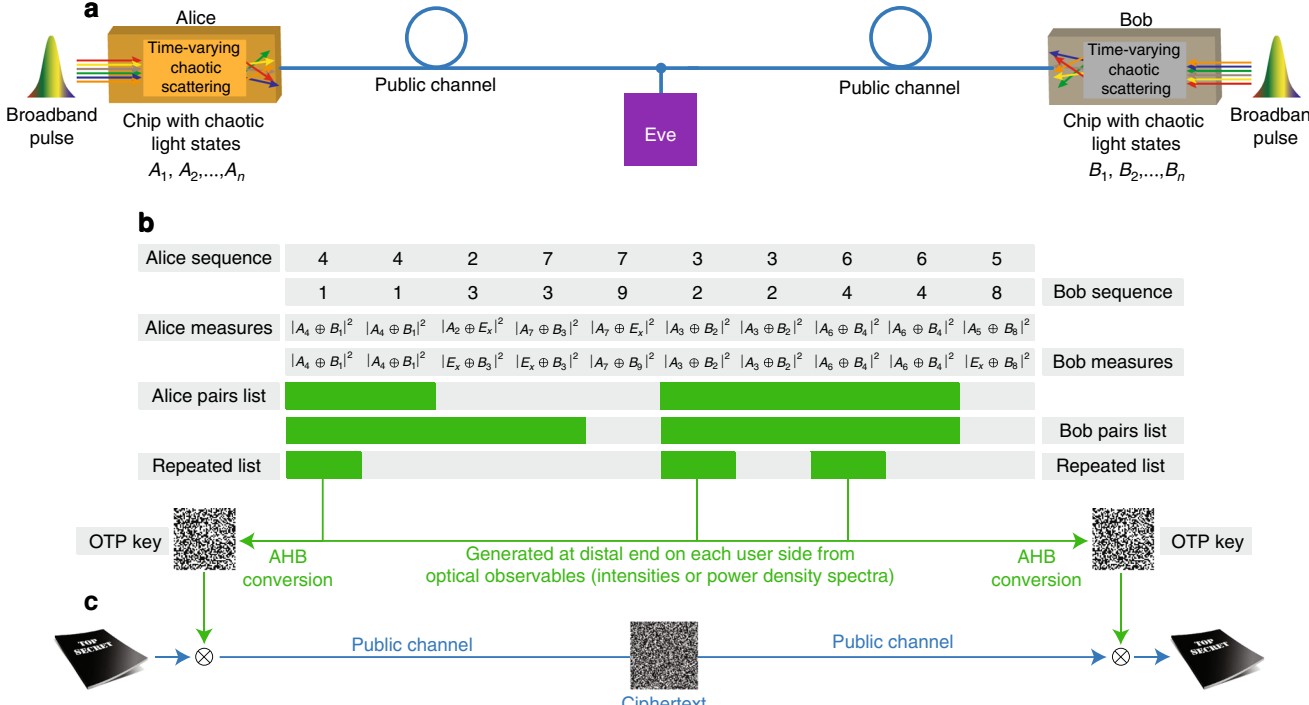

**Fig. 1 Protocol scheme for perfect secrecy key generation on classical channels. a** Communication setup on a classical public optical channel with the users (Alice and Bob) and attacker (Eve). Alice and Bob possess two different chips that generate chaotic light states $A_n$ and $B_{n'}$. **b** Communication and key generation steps: Alice and Bob launch broadband pulses from their sides and transmit different chaotic states $A_n$, $B_{n'}$, always measuring correlated mixed chaotic states when Eve does not actively interfere on the channel with additional states $E_x$. States $A_n$ and $B_{n'}$ are independently randomly chosen by Alice and Bob. At the end of the transmission, Alice and Bob generate a key from the sequence of overlapping repeated sequences with the adaptive high boost (AHB) transform. **c** Encryption and decryption scheme via bitwise XOR between the text and the generated key.

time modulation,...) arbitrarily chosen by Alice and Bob. Alice and Bob's pulses are not required to be identical: their differences constitute the main source of uncertainty and set the desired communication bit error rate (BER) in the communication. The scattering system satisfies the following four conditions: (1) the scattering process inside each chip is fully chaotic[30]: any launching condition follows a chaotic dynamics; (2) Alice and Bob's chips are in thermodynamic equilibrium with the environment, with no structural change appreciable for light propagation during the communication of each state; (3) any modification to the distribution of scatterers leads to a new chaotic system with exponentially diverging trajectories with respect to the previous one; (4) the chips are structurally modified in time before and after each communication by two physical irreversible processes (e.g., deformations, addition of scatterers, etc.), chosen and applied independently by Alice and Bob, creating a new chaotic scattering system in which all trajectories are exponentially different from the previous one.

The use of static light scatterers in information security is introduced in ref. [31] and offers computational and probable security in both authentication problems and cryptographic key generation, providing advantages over electronic schemes in terms of volumetric physical data storage versus standard electronic databases[32–41]. The security of schemes based on complex scattering structures relies on two conditions: (i) the physical scattering object is kept secret to an adversary, (ii) the assumption that this structure cannot be cloned. These arguments do not offer unconditional security and are subject to the same security concerns of electronic schemes. While recent work demonstrated that it is indeed experimentally possible to clone a physically unclonable function[42], perfect secrecy requires proving the system security in the limit where the adversary accesses the system before or after the communication, copying all the system's parts. In this work, as we are discussing the limit of perfect secrecy, we remove conditions (i)–(ii) and we consider an adversary with the technology to clone any type of scatterer.

During each step of the communication, Alice and Bob randomly choose an input condition $n$ and $n'$, respectively, send light signals, and measure the output (Fig. 1b). After Alice and Bob choose randomly either to keep the launching condition or to change it, the process is then repeated.

Due to the reciprocity[43] of the communication network connecting Alice and Bob, if Eve does not perform active eavesdropping, the users measure identical optical observables (intensity, power density spectra (PDS), etc.). For instance, when Alice sends a chaotic wavepacket $A_n$ to Bob, he measures an optical observable, for example, the intensity $|A_n \oplus B_{n'}|^2$ associated with the combined light state $A_n \oplus B_{n'}$ ($\oplus$ is the operator that combines the states after the propagation over the channel). When Alice measures the output at her end, she measures the reciprocal state $B_{n'} \oplus A_n$ with an identical observable $|A_n \oplus B_{n'}|^2 = |B_{n'} \oplus A_n|^2$.

At the conclusion of the sequence, Alice and Bob communicate on the public line all cases of the acquired data that did not change, extracting an OTP key from overlapping repeated sequences. The key is generated by converting the exchanged intensities or PDS into binary sequences with the adaptive high boost (AHB) technique[44]. The key is then used at each user side to encode and decode data via bitwise XOR, following the Vernam cipher (Fig. 1c).

The communication protocol described in Fig. 1 can be regarded as a classical version of the original BB84 QKD scheme developed by Bennett and Brassard[45], in which the scatterers act as generator

of random states, and the reciprocal communication line provides correlated measured states to the users Alice and Bob.

At variance with classical schemes based on complex scatterers[40,41], the protocol presented here does not require first encounter or initial secure communication among the users (apart from authentication), thus providing a classical alternative to QKD.

In the quantum limit, when a user (say Bob) launches a single photon in the chip, the receiver (Alice) measures a photon emerging at a random position from the chip. If Alice injects the photon back in the same scattering channel, the reciprocity theorem of quantum mechanics[46] guarantees that Bob measures the emerging photon in the same input channel he originally used. This process shares some similarities to the quantum BB84 scheme, with scattering channels playing the role of random polarization states. However, there are also differences. For the users to exchange the same sequence of bits, they need to initially agree on a common dictionary that associates the same string of bit to correlated input–output positions in Alice and Bob's chips. This operation is not required in the classical limit (as the users measure the outcome of large ensemble of photons on all channels) and in the BB84 scheme.

**Perfect secrecy of the cipher.** The Vernam cipher has the perfect secret property if: (i) the key exchanged is as long as the message, (ii) each key is used only once and is uncorrelated to the new one, and (iii) the key is known only by the users. The scheme of Fig. 1a exchanges keys of arbitrary length on a classical optical channel at full speed. It therefore offers a viable implementation of the first requirement.

The covariance matrix $K_{nn'} = \langle A_n A_{n'} \rangle$ of the correlation among Alice chaotic wavepackets, with $\langle ... \rangle$ denoting averaging over amplitudes and phases, is a delta function:

$$K_{nn'} = \left\langle \sum_{mm'} a_{nm} a_{n'm'} \int dt \cos(\omega_m t + \phi_{nm}) \cos(\omega_m t + \phi_{n'm'}) \right\rangle, \quad (2)$$
$$= \sigma_n^2 \delta_{nn'},$$

with $\sigma_n^2 = K_{nn}$. Equation (2) arises from the fact that amplitudes are uncorrelated, with $\langle a_{nm} a_{n'm'} \rangle = 0$ for $n \neq n'$ and $m \neq m'$. The same condition holds at Bob's end, with $B_{nn'} = \langle B_n B_{n'} \rangle = \chi_n^2 \delta_{nn'}$. This implies that both Alice and Bob states are uncorrelated, and keys generated from combined states $A_n \oplus B_n$ are also uncorrelated. Therefore, the protocol of Fig. 1 satisfies the second requirement.

**Perfect secrecy of the key distribution.** We now consider the ideal case (Kerckhoff principle) in which the system falls in the hands of the adversary, who knows all the details of the enciphering/deciphering process and has access to the ciphertext. The only unknowns are the key and the input conditions (including the arbitrary chosen transformations) of the users.

As the system is classical, Eve can store all the signals launched by Alice and Bob and then she can attempt a search on each user's chip for the input conditions that generated the states she measured. Once Eve knows the input conditions, she can launch the same states and recreate the key.

The second law of thermodynamics prevents this attack. Every time Alice and Bob change the chip with an irreversible process, they increase the total entropy of the system and the environment, creating new chaotic structures exponentially different from the ones used in the communication (conditions 3 and 4). If Eve accesses the system, it is impossible to recreate the initial chips and to perform any search, as this requires reverting the

transformation of Alice and Bob with an entropy decrease, thus violating the second law.

Another possibility is to make an identical copy of the system in all its parts, and to attempt the search at the next communication. This is a major vulnerability of all current classical cryptographic schemes. The system presented in this work, on the contrary and due to the use of irreversible thermodynamic transformations, is protected by such attack. The search task, in fact, requires Eve to generate the same chaotic scatterers as of Alice and Bob's, so that their transformations are cloned prior to the communication. As the chips are in equilibrium with the environment (point 2), this task requires replicating the surroundings of Alice and Bob's chips. This condition is essential for enabling Eve's copied chips to reach the same equilibrium state of the original chips of Alice and Bob. The second law of thermodynamics makes this operation not physically possible. Eve, in fact, cannot replicate the exact time at which Alice and Bob perform their transformations. If Eve does the transformation after Alice or Bob, the environment will be different, as it existed at least one irreversible transformation in time (the one of Alice or Bob) that increased its entropy, and vice versa if Eve performs the transformation before the users. It will be therefore impossible for Eve to clone the transformation of Alice and Bob. Due to points 3 and 4, Eve will generate new chaotic scatters that are exponentially different from the ones that Alice or Bob are using and, as such, useless.

This leaves to the attacker just one possibility: extract the key from the information available on the system.

In any possible attack, the data available to an attacker are the observables related to the chaotic wavepackets $A_n$ and $B_{n'}$ transmitted. Eve can measure these states and attempt to reconstruct Alice and Bob's key. In the analysis below, we demonstrate that the outcome is always a key in which each bit has 50% probability of being correct and 50% probability of being wrong, regardless of the type of attack. This implies perfect secrecy[6]: a posteriori probabilities of Eve's key representing Alice and Bob's key is identically the same as the a priori probability of guessing Alice and Bob's key before the interception.

In the scheme of Fig. 1a, the solely experimental observables are noninstantaneous quantities in frequency or time, such as PDS or intensities, while instantaneous values of amplitude and phase are of traveling photons, which are not observable. This limitation also applies to interferometric detection and time gating[47–49], which require periodic signals in time or precise knowledge in advance of the pulse's properties. To measure the instantaneous state of a randomly generated chaotic wavepacket $A_n$ or $B_{n'}$ that is never replicated, the only possibility is to accelerate electrons at relativistic speed in order to follow the dynamics of photons, but this requires an infinite amount of energy[50].

**Security against time-domain attacks**. We analyzed the limit in which Eve developed a technology to access instantaneous values of intensity, and considered a scenario independent from the source and channel used, which are set to mathematical Dirac delta $\delta(t)$. In this case, the intensity of the combined state $I_{A_n B_{n'}}(t) = |A_n \oplus B_{n'}|^2 = |A_n(t) \otimes B_{n'}(t)|^2$, with $\otimes$ the convolution operator.

During each step of the communication, Eve stores the intensities $|A_n(t)|^2$ and $|B_{n'}(t)|^2$ of transmitted states, and attempt the reconstruction of the state $|A_n \otimes B_{n'}|^2$ by combining the states at disposal via $|A_n|^2 \oplus |B_{n'}|^2$, with $\oplus$ a chosen operator. We here considered all the possible cases of $\oplus = +, \cdot, \otimes$ (sum, product, convolution).

The outcome of this attack is quantified by the average Shannon information entropy contained in each bit measured by Eve, and calculated from the average Shannon information entropy $H = -d\log_2 d - (1-d)\log_2(1-d)$ per bit, with $d$ the difference in bits between the key of one user (Alice) and the key reconstructed by Eve. The Shannon entropy $H$ quantifies the uncertainty of Eve for every bit measured. When $d = 0$ or $d = 1$ (Eve measures the same or the opposite of Alice), the information entropy of Eve is zero, because Eve predicts the key with no uncertainty. In the other cases, $H$ is a positive function with maximum of $H = 1$. In this condition, Eve has 1 bit of uncertainty for every bit measured and zero information on the key.

Figure 2a shows the average uncertainty of Eve when Alice and Bob mix random wavepackets described by Eq. (1) and containing an increasing number of different frequencies $\omega_1, ..., \omega_M$. Computational details are furnished in Supplementary Note I. A chaotic wave, arising from chaotic scattering, obeys a universal Gaussian statistics for the intensity[29,30] $P(I) = \alpha e^{-\alpha I}$ ($I = A_n^2$ for Alice and $I = B_{n'}^2$ for Bob), and it is attained at large $M$, where the states $A_n$ and $B_{n'}$ are completely randomized. At lower $M$, the wavepackets $A_n$ and $B_{n'}$ are aperiodic superpositions of waves with no general behavior.

In the limit of small $M$, the statistics of the wavepackets are different, and Eve's uncertainty oscillates in a large interval with situations in which the attacker can infer the key by combining her measurements via $\otimes$ (Fig. 2b). Using different operators, as intuitively expected, yields no information. When the number $M$ of frequencies increases and the states $A_n$ and $B_{n'}$ become chaotic, each realization shows the same universal features (Fig. 2c) and the variance of the uncertainty collapse (Fig. 2a). In this limit, the uncertainty of Eve becomes unitary ($H \geq 0.998 \pm 0.01$): the information accessible is not sufficient to reconstruct the complex state being formed at Alice and Bob's end and the system is unconditionally secure.

Active attacks (see Supplementary Note II) introduce deterministic errors in the communication sequence between Alice and Bob with no information for Eve. Errors are small and scale as $1/N$, with $N \gg 1$ being the number of chaotic states available in Alice and Bob's chips. These errors can also be eliminated by using information reconciliation and privacy amplification[51–54], both conducted over the public authenticated channel. With information reconciliation, Alice and Bob obtain an identical key at each user's side by the exchange of minimal information (such as the mere bit parity of block key sequences). The second phase, privacy amplification, is then applied to eliminate effectively the information acquired by Eve during the reconciliation step. Privacy amplification is typically performed by using universal hash functions, which generate a new shorter key, on which Eve has zero information.

**Security against spectral attacks**. The transfer function $H(\omega)$ of the system connecting Alice and Bob is represented as follows:

$$H(\omega) = S(\omega) \cdot H_A^{(n)}(\omega) \cdot H_B^{(n)}(\omega) \cdot C(\omega) \cdot \alpha(\omega), \qquad (3)$$

with $H_A^{(n)}(\omega)$, $H_B^{(n)}(\omega)$ the transfer function of the chip of Alice and Bob, respectively, $C(\omega)$ the contribution of the transmission system, $S(\omega)$ the spectrum of the input source, and $\alpha(\omega)$ the coupling coefficient with spectrum analyzer, with $|\alpha(\omega)| \leq 1$ for energy conservation. For reciprocity, the coefficient $\alpha(\omega)$ is the same for both users. The random PDS $P_A = |H_A^{(n)}(\omega)|^2$ and $P_B = |H_A^{(n)}(\omega)|^2$ change at every transmission step, due to the different input conditions selected by each user.

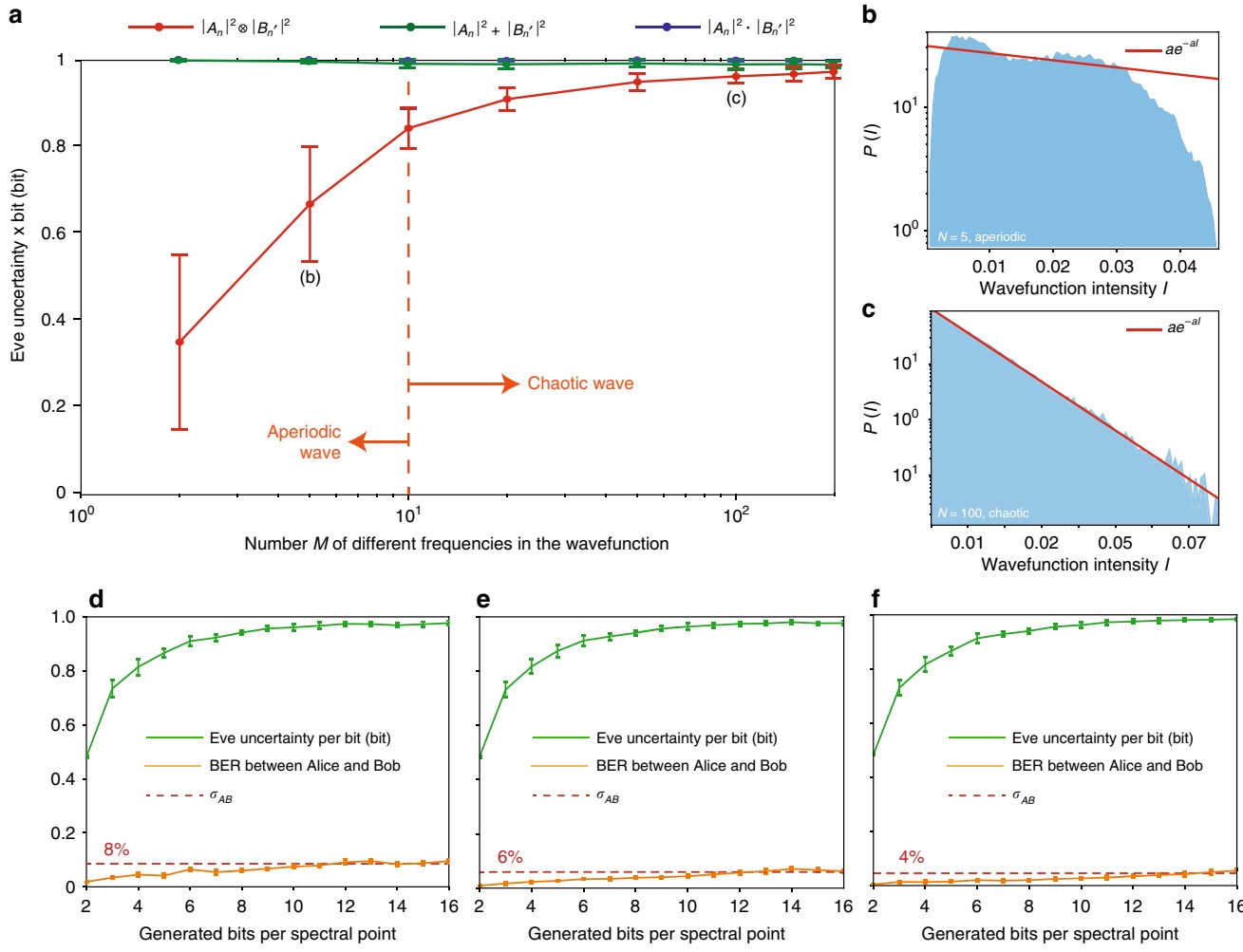

**Fig. 2 Protocol security against time-domain and spectral attacks. a** Uncertainty per bit measured by an ideal attacker for all possible types of attempted time-domain key reconstruction: $\otimes$ (red line), $+$ (green line), $\cdot$ (blue line), when the users mix random wavepackets with increasing number $M$ of different frequencies. **b**, **c** Statistics of intensity (solid blue area) of single wavepackets for **b** aperiodic and **c** chaotic cases, versus universal Gaussian statistics (red line). **d**–**f** Results of spectral attacks: Eve's uncertainty per bit (solid green line), bit error rate (BER) between Alice and Bob keys (solid red line) for different standard deviations $\sigma_{AB}$ between the power density spectra of the combined states measured by Alice and Bob, as a function of the number of bit $N_b$ extracted for each spectral point measured. All panels report average values (solid lines) and standard deviations (error bars).

In a spectral attack, Eve measures the spectra $P_{EA}$ and $P_{EB}$ transmitted over the communication line with identical copies of the users' spectrum analyzers, and then attempts a reconstruction of the combined state via $P_{EA}P_{EB}/P_S$, with $P_S$ the spectrum of the source. In the best ideal scenario for Eve, this operation returns the following estimate (Supplementary Note III):

$$\frac{P_{EA} \cdot P_{EB}}{P_S} = \langle P_{\text{Bob}} \rangle \frac{1}{|\alpha|^2} + \sqrt{3} \frac{\Delta}{|\alpha|^2}, \quad (4)$$

with $\langle P_{\text{Bob}} \rangle$ the mean value of the combined state $P_{\text{Bob}}$ measured by Bob and $\Delta$ the uncertainty measured by Alice and Bob in their combined states. We analyze the security of the system in the worst case for the users, in which $|\alpha(\omega)|^2 = 1$. This limit is practically impossible to achieve: it implies a technology that can measure a state without perturbing it, and this would at least violate the projection postulate of quantum mechanics.

We considered different communications scenario, in which Alice and Bob measure correlated spectra with different statistical fluctuations, and developed a multi-bit AHB transform (detailed in Supplementary Note IV), which optimizes the extraction of information from acquired users spectra. Commercially available detectors furnish at least 16 bits for each spectral point: if this

information is maximized, the key generation workload is highly reduced.

In the security analysis, we considered statistical fluctuations between Alice and Bob PDS with standard deviation $\sigma_{AB} \leq 8\%$, which is typically met in classical communication networks. We set the maximum tolerable BER equal to the statistical fluctuations in the spectra $\sigma_{AB}$. This implies that the BER is lower than the tolerable limit of 11% set for QKD[7].

Figure 2d–f shows the outcomes in terms of BER and adversary uncertainty per bit resulting from a spectral attack for different $\sigma_{AB}$. Results are calculated from a statistical set of $10^6$ different chaotic PDS measured by Alice $P_{\text{Alice}}$, Bob $P_{\text{Bob}} = P_{\text{Alice}} + \Delta$ and reconstructed by Eve $P_E = P_{\text{Alice}} + \Delta'$ at the theoretical limit, with: $\langle \Delta \rangle = \langle \Delta' \rangle = 0$, $\sqrt{\langle \Delta^2 \rangle} = \sigma_{AB}$, $\sqrt{\langle \Delta'^2 \rangle} = \sqrt{3}\sigma_{AB}$.

Figure 2d–f demonstrates that independently from the communications scenario considered, when the BER reaches $\sigma_{AB}$, the users can distill an OTP key with $N_b > 10$ bits per spectral point, while maintaining a unitary uncertainty to the attacker (uncertainty per bit higher than 0.99 bit). As in Fig. 2a, the variance in Eve's uncertainty is negligible. This arises from the use chaotic spectra with universal statistics: the outcomes are independent on the particular sequence considered.

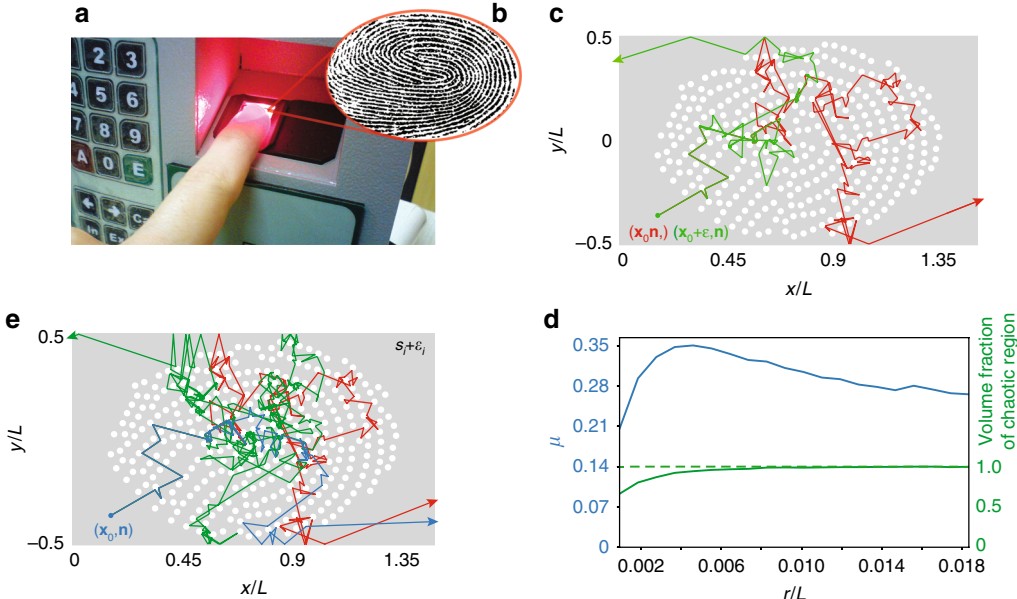

**Fig. 3 Integrated fingerprint silicon chips design and chaotic analysis. a, b** Biometric fingerprint acquisition and transformation into a chaotic resonator (**c**). **c** Dynamics of two trajectories (red and green lines) obtained from two input conditions with identical launching direction **n** and position $\mathbf{x}_0$ displaced by the smallest number $\epsilon$ representable at the computer. **d** Averaged Lyapunov exponent $\langle \mu \rangle$ in the phase space of possible input conditions (blue line) and percent of the phase space filled with chaotic dynamics for different radius $r$ of the scatterers (green line). **e** Dynamics of three identical input conditions (red, green, and blue lines) in three infinitesimally different resonators obtained by randomly shifting the position of each scatterer by $\epsilon$ in each spatial direction. The image in **a** is adapted from ref. [65] under license CC BY-SA 4.0.

If we set $N_b$ to the point where the BER equals $\sigma_{AB}$, $N_b$ increases from (d) $N_b = 11$ to (e) $N_b = 12$, and then to (f) $N_b = 14$, and can be controlled by acting on $\sigma_{AB}$.

These results demonstrate that at the theoretical limit and in the worst case for the users, the attacker has zero information on the key and the system is unconditionally secure.

Supplementary Note V demonstrates that in the absence of active eavesdropping and by using the multi-bit AHB transform, the number of pulses $N_p$ to be transmitted for generating an OTP key can be reduced to $\approx 1/1000$ of the length of the message.

**Physical implementation**. We developed chaotic chips from human fingerprints. After biometric scanning (Fig. 3a, b), the digital fingerprint image is transformed into a chaotic micro-resonator composed by a series of point scatterers made by reflective nanodisks of constant radius $r$ (Fig. 3c). The micro-resonator acts as a chaotic billiard[55–60]. In Figure 3c–e we optimized the resonator chaos against conditions 1 and 3 from the communication protocol (Supplementary Note VI).

The fingerprint resonator is characterized by a large number of disjoint convex bodies, representing a finite version of the Lorentz gas billiard, which is known to possess a strongly chaotic behavior[55]. Figure 3c shows the propagation of light rays of two input conditions $(\mathbf{x}_0, \mathbf{n})$ (solid red line) and $(\mathbf{x}_0 + \epsilon, \mathbf{n})$ (solid green line), having the same initial orientation **n** and spatially displaced by a random vector $\epsilon$ with $|\epsilon| = \epsilon_{min} = 2.2 \times 10^{-16}$, the smallest floating point number representable at the computer. After few collisions, the dynamics diverges exponentially.

Figure 3d calculates the average Lyapunov exponent $\langle \mu \rangle$, which quantifies the average exponential grow of different input conditions, and the volume of phase space of input conditions filled by chaotic dynamics as a function of the scatterers' radius $r$ (Supplementary Note VI). The designed fingerprint resonator with $n_s = 322$ scatterers is fully chaotic (as requested at point 1) for $r \geq 0.008L$, with $L$ being the resonator width along $y$.

Figure 3e analyzes the fingerprint structure against conditions 3 and 4. The plot shows the dynamics of three identical input conditions (solid blue, red, and green lines) launched in three different fingerprint resonators implemented by randomly shifting the positions of each scatterer $\mathbf{s}_i$ by a vector with random orientation $\epsilon_i$, with $|\epsilon_i| = \epsilon_{min}$. An infinitesimal transformation in the fingerprint's scatterers leads to a chaotic structure with exponentially diverging trajectories with respect to the old one. This is a general result, demonstrated in Supplementary Note VII, along with other possible transformations that satisfy requirements 3 and 4.

Figure 4a shows the typical dynamics of light in a fully chaotic fingerprint resonator with $L = 7$ μm and $r = 0.012L$, calculated from finite-difference time-domain simulations for a structure of air holes on a Silicon substrate excited by a TE polarized, 150-fs-long pulse centered at $\lambda = 1550$ nm. Chaotic scattering randomizes the wavefront, generating wavepackets with universal Gaussian statistics (Fig. 4b).

Figure 4a–d analyzes the correlations among chaotic states extracted from transmitted PDS. Spectra are advantageous over intensity signals, favoring less expensive implementations. We calculated transmitted electromagnetic spectra for both TE and TM polarized point sources of 150 fs duration, centered at the wavelength $\lambda = 1550$ nm, and launched at $x = y = 0$ with displacements $y_1, y_2, \ldots$ along $y$ within 1 μm range with 20 nm resolution. For each input position, we computed the transmitted energy spectrum and transformed it into a binary sequence by the AHB technique. We then computed the entropy correlation matrix $\mathcal{H}$, with elements $\mathcal{H}_{ij} = -d_{ij} \log_2 d_{ij} - (1 - d_{ij})\log_2(1 - d_{ij})$ being the Shannon information entropy of the hamming distance $d_{ij}$ among the bit sequences $i$ and $j$ arising from PDS obtained by input shifts $y_i$ and $y_j$. The entropy correlation matrix is strongly diagonal (Fig. 4c), showing that the generated bit sequences are completely uncorrelated. A displacement beyond 200 nm provides uncorrelated sequences (Fig. 4d).

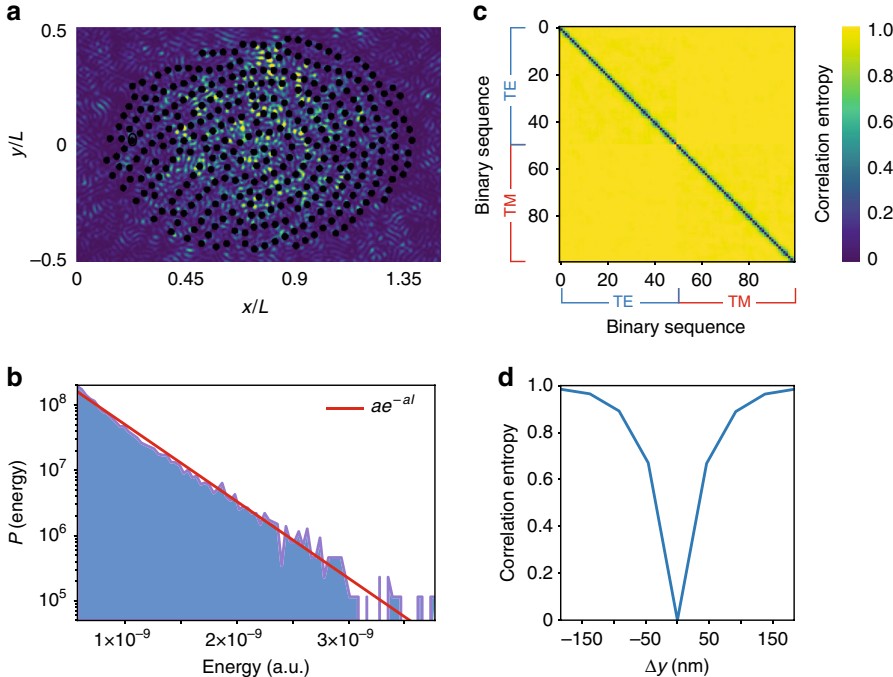

**Fig. 4 Wave analysis of fingerprint chips from finite-difference time-domain (FDTD) simulations. a** Electromagnetic energy distribution in a resonator made by air pillars on a silicon substrate for an input pulse at $\lambda = 1550$ nm and 150 s long. **b** Probability distribution of the electromagnetic energy inside the resonator (solid blue area) versus universal Gaussian statistic (solid red line). **c** Correlation entropy between the binary sequences generated from transmitted electromagnetic spectra obtained by shifting the launching position of the input beam within 1 μm from $x = y = 0$ with displacements of 20 nm for TE and TM polarized excitation. **d** Average correlation entropy for displacements within ±150 nm.

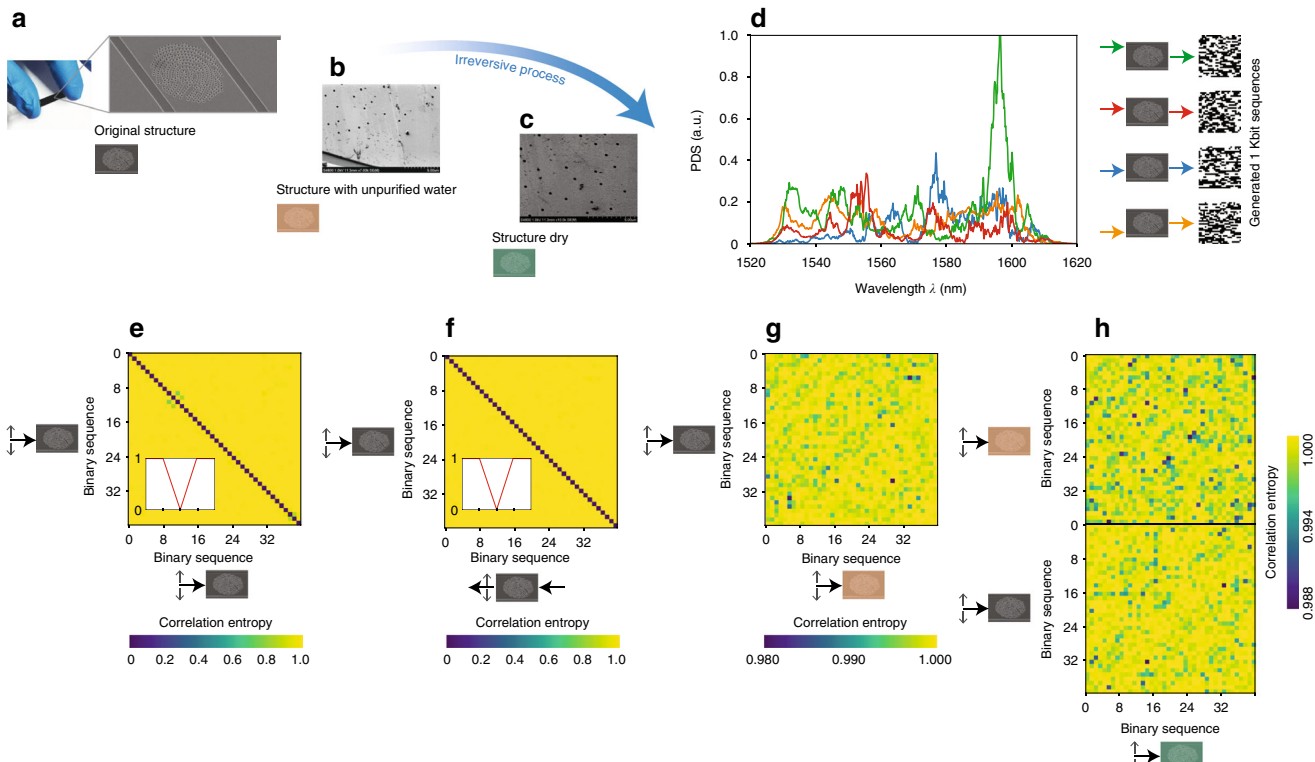

**Fig. 5 Experiments on single fingerprint chips. a** Scanning electron microscope (SEM) image of a fabricated sample. **b, c** Example of irreversible transformation sequence, characterized by **b** the deposition of water not purified with colloidal suspensions, and **c** same chip dried out. **d** Transmission spectra and generated bit sequences corresponding to four different input positions, spaced by 10 μm each. **e–h** Correlation entropy between the bit sequences generated from identical input positions in **e** the same chip, **f** the same chip in direct and reciprocal configuration, **g** the original chip and the chip modified in **b**, **h** the chip modified in **b**, and the chip modified in **c** and the original structure.

In the system of Fig. 1, by using displacement and two polarizations, we can generate $(1000/0.2 \times 2)^2 = 10^8$ different spectra for every millimeter of chip length $L$. If we measure each spectrum with 1024 pixel and convert each pixel into $N_b$ bits, we can generate $0.1 \cdot N_b$ Tbit of different keys for every millimeter of chip unit length $L$, and for every communication.

In the following experimental demonstration, fingerprint chips are fabricated by e-beam lithography, patterning the fingerprint resonator structure onto a silicon substrate developed with a silicon-on-insulator platform that is CMOS (complementary metal–oxide–semiconductor) compatible and operating in the standard telecommunication C + L band[47]. Figure 5a shows a scanning electron microscope of a fingerprint chip. The structure is $L = 40\,\mu m$ wide and $100\,\mu m$ long.

Figure 5b, c shows an example of an irreversible process[61] that can be applied to the chips. It consists in first depositing on top of the sample a non-purified drop of water, which naturally contains colloidal occlusions that act as additional scatterers (Fig. 5b). When the chip dries out naturally, we obtain another distribution

of impurities on the chip's surface (Fig. 5c). In commercial applications, this step can be accomplished by solid-state scatterers, such as, for example, doped hydrogels, which are dynamically deformable[62–64].

To characterize the chips in (a–c), we launched light signals of 100 nm bandwidth, centered around the communication wavelength 1550 nm, by end fire coupling with a ×60 aspheric lens, mounted on a XYZ stage with repeatable spatial shifts of 0.5 μm.

Figure 5d shows transmission optical spectra recorded at different input positions, indicated on the right, and the corresponding generated bit sequences, each of 1024 bits, for the original chip (a). Figure 5e shows the entropy correlation among 40 different input positions shifted by 1 μm each. In agreement with the theoretical predictions of Fig. 4c, d, generated bit sequences are uncorrelated with each other. Figure 5f verifies the optical reciprocal behavior of the chip against condition 2. The figure shows the correlation among bit sequences created by launching signals from the input and collecting transmission spectra at the output, versus the bit sequences measured by

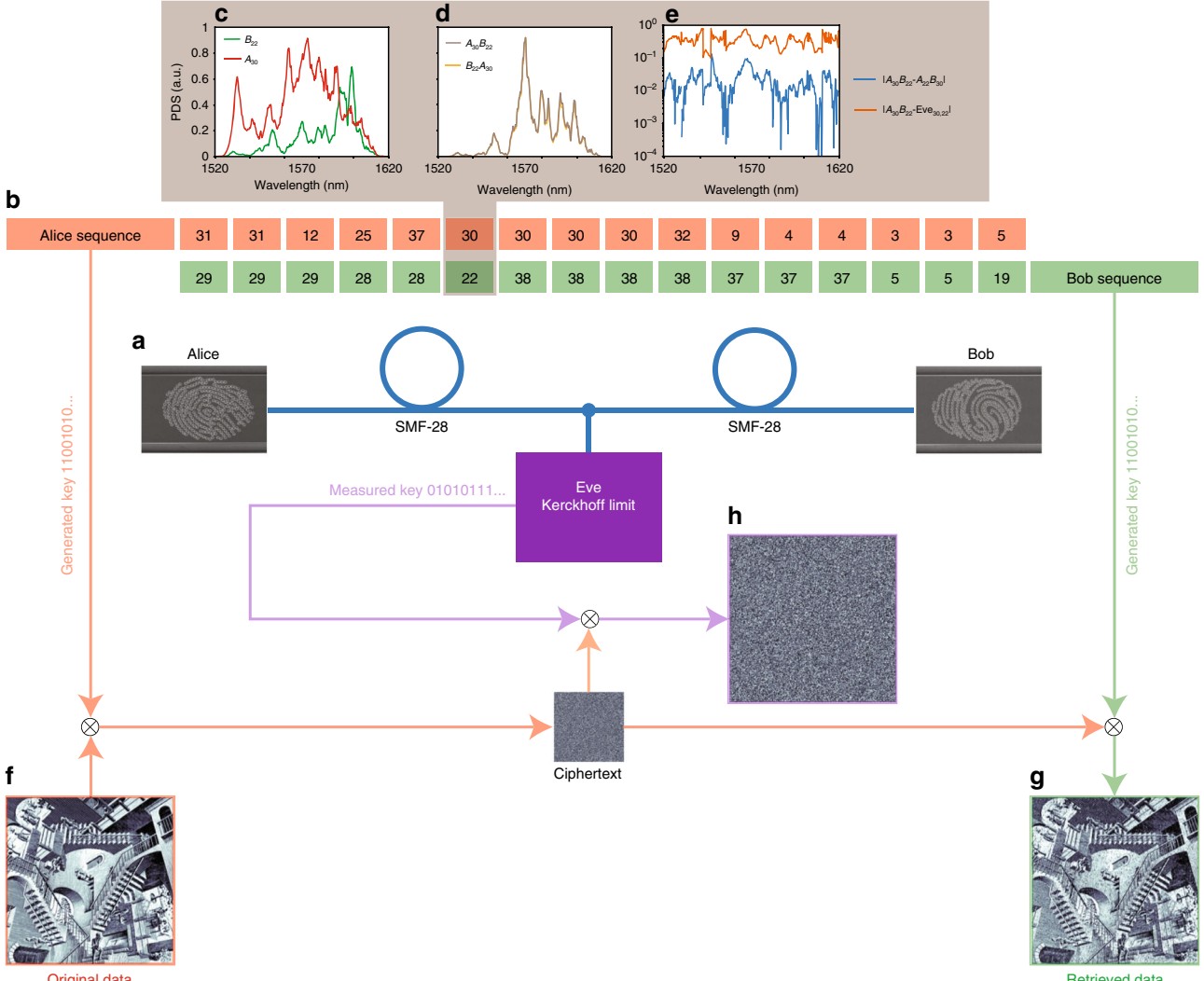

**Fig. 6 Experiments on key distribution and attacks in the spectral domain. a** Experimental setup composed by two chips connected by a standard monomodal optical fiber SMF-28. **b** Communication sequence of the chosen input conditions between Alice and Bob. **c–e** Example of data sent during one communication step, including **c** the individual random spectra of the user as measured from the communication line, **d** the combined spectra at the distal end measured by Alice and Bob, **e** the absolute spectral differences in the combined states measured by the users (solid blue line) and the state reconstructed by an ideal attacker (solid orange line). **f–h** Encryption and decryption experiments. **h** shows the results of Eve from the best possible attempt to recreate the key of the users.

launching signals from the output and collecting spectra at the input. The chips have correlated bit sequences obtained in direct and reciprocal launching conditions (Fig. 5f, diagonal), and uncorrelated to each other (Fig. 5f, yellow area).

Figure 5g analyzes the entropy correlation between the bit sequences created from the same positions in the chips at (a) and (b), before and after the transformation, respectively, while Fig. 5h shows the correlations among sequences generated from chips at (a), (b), and (c). The sequences are uncorrelated with each other, with an average correlation entropy per bit of $\langle \mathcal{H} \rangle = 0.998 \pm 0.001$ bit, experimentally demonstrating that the irreversible transformation (a, b, c) generated different chip responses.

Figure 5g, h experimentally proves that the chips satisfy the conditions 3 and 4 of the communication protocol.

**Cryptographic transmission**. We used the setup of Fig. 6a, assembled with inexpensive classical optical components. We employed a standard single mode telecommunication fiber SMF-28 that connects two fully chaotic fingerprint chips with $L = 40 \, \mu m$. We chose different input conditions by using XYZ translation stages with $1 \, \mu m$ spatial shifts. Supplementary Note VIII discusses an integrated structure with on-chip coupling from end to end of the communication line. As in the previous experiments, we employed light pulses with 100 nm bandwidth and measured PDS with 0.1 nm resolution.

Figure 6b shows a typical communication with a list of input positions selected by each user. Figure 6c, d analyzes the transmission for one set of input conditions showing: (c) the PDS sent by Alice and Bob and (d) the spectra measured at the distal end by the users. The PDS of the combined state $PDS_{B_{22}A_{30}}$ measured by Alice is correlated to the spectrum $PDS_{A_{30}B_{22}}$ measured by Bob. The solid blue line in Fig. 6e quantifies the difference between the data measured by Alice and Bob.

The solid orange line in Fig. 6e shows the difference between the spectrum measured at Bob's end and the one reconstructed by Eve with a spectral attack, by measuring transmitted spectra $P_{B_{22}}$, $P_{A_{30}}$ and combining them via $P_{B_{22}}P_{A_{30}}/P_S$, with $P_S$ the spectrum of the source (Supplementary Fig. 1). All Eve's measurements are performed with the users's spectrum analyzers, without using any component that is not present in the setup. Even in this scenario, there are significant differences between Bob and Eve's spectra.

Figure 6f–h visualizes these results with encryption and decryption experiments. Alice encodes the data being sent out (f) with a key generated from the sequence of repeated spectra with the multi-bit AHB transform optimized for $\sigma_{AB} = 7\%$, which is the average standard deviation generated in the system of Fig. 6a.

The ciphertext generated is then transmitted to Bob end, who decodes it with his own generated key. The image decoded by Bob is correctly retrieved from the original (Fig. 6g). Conversely, the image decoded by Eve is just white noise. With this elementary setup and with the chips designed ($L = 40 \, \mu m$), we can extract $40^2 \cdot N_b \cdot 1024 = 2 \cdot N_b$ Mbit of different keys at each communication.

In Supplementary Fig. 2 we report the results of the NIST statistical test suite applied to the keys generated with the system of Fig. 6. The key generated passed all the tests, validating the scheme against the NIST standards for real-world applications.

**Discussion**. We have demonstrated a protocol for a perfect secrecy cryptography that uses CMOS-compatible fingerprint silicon chips, which transmit information on a public classical optical network. The system's security is evaluated following the Kerckhoff principle. The second law of thermodynamics and the exponential sensitivity of chaos prevents the attacker from getting any information on the key being exchanged by the users. The protocol proposed is fully compatible with the techniques of privacy amplification and information reconciliation already developed for QKD. Beyond the initial communication required for authenticating the users, the system does not require electronic databases, private keys, or confidential communications. Combined with the technological maturity, speed, and scalability of classic optical communications, the results show a open pathway towards implementing perfect secrecy cryptography at the global scale with contained costs.

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

## Acknowledgements

A.D.F. acknowledges support from EPSRC (EP/L017008/1). A.F. acknowledges support from KAUST (OSR-2016-CRG5-2995). The research data underpinning this publication can be accessed at https://doi.org/10.17630/19156fc3-cc1f-4ee3-b553-f02042cf89a0.

## Author contributions

A.F. directed the work and developed the main theory. A.D.F. fabricated and measured the samples. V.M. helped on the experiments and contributed to the design of the communication protocol. A.C. provided conceptual comments. A.F. wrote the manuscript, with equal input from all authors.

## Competing interests

A provisional patent application (US Patent USPTO16132017) has been filed on the subject of this work on which all of the authors are listed as inventors.
