## [Peer Review File · Nature Communications]

Reviewers' comments:

Reviewer #1 (Remarks to the Author):

In the manuscript "Perfect secrecy cryptography via correlated mixing of chaotic waves in irreversible time-varying silicon chips," the authors demonstrate key distribution that provides perfect secrecy on a classical communication channel. They utilize chaotic silicon integrated photonic chips to act as a unique, random, and irreversibly modifiable filters for the ultrafast optical pulse. The filter behavior, which is dynamically changed in time by the user, serves as the source of the shared key between the two parties. This shared key can then be used to encrypt a message using the one-time pad (OTP) approach. Notably, in contrast to quantum key distribution (QKD), the authors' approach utilizes a classical signal.

The main novelty of the manuscript is the ability to distribute a shared key for perfect secrecy communications using classical signals. This is a very exciting possibility: to be able to do key distribution using classical means and provide perfect secrecy has not been accomplished before. This is extremely exciting compared to the only proven alternative of a quantum approach. However, the manuscript is extremely confusing and needs a great deal of simplification in order to provide the reader with a clear picture of the technique and how it is able to achieve this perfect secrecy.

The silicon integrated photonic devices utilized as the source of a chaotic filter are incremental, and are a forced recreation of the free-space scattering optical PUFs previously demonstrated in the literature, and therefore suffer from many of the same shortcomings as their free-space counterparts; chief among them being the need for free-space alignment due to their multimode nature. The research presented in the manuscript is novel and interesting, however it is not suitable for publication at this time in its current form. The manuscript is bogged down with extraneous details, and the crux of the functionality of the system is hidden in the details of the supplementary files. I had to read the paper several times before I could understand how the system works. After the reader finally is able to understand how the key is generated, the next most obvious train of thought is to question why the system is secret. The most natural way to attack the system is to have an eavesdropper passively measure the transmissions and attempt to recreate the key. This most basic attack is skipped over in the manuscript, and after the authors address many ancillary attack approaches, they finally direct the reader to the supplementary files to try to understand the crux of how the key creation can remain secret. This will be addressed in more detail below.

The most obvious way to attack the proposed system would be for an eavesdropper to passively eavesdrop and attempt to measure the signal. The shared key is generated through the combined response of two chaotic chips, and due to reciprocity, the combined response is the same regardless of what direction, therefore both users, Alice & Bob, can measure the same response from light originating from the opposite user's location. The essential problem to provide security is ensuring that an eavesdropper (Eve) cannot also measure that combined response. In this case, the reason is because the key extraction is in essence made very sensitive to the SNR of the measurement, and each of the users have a better SNR than the eavesdropper. This means that the transmission must be performed with the proper amount of injected noise such that the users can extract the key without error, whereas no information about the key is leaked to Eve. This (my last two sentences) is the crux of the manuscript, and it is buried in the details of the supplementary file. It is completely absent from the figures of the manuscript, and should be included in the overall system figure

(figure 1).

Many aspects of the system are very similar to previously published work and the authors do not put their work in the context of the existing literature (nor are the two very relevant works referenced). Secure communications on classical optical channels have been demonstrated before, with the system in "Physical key-protected one-time pad," by Roarke Horstmeyer, Benjamin Judkewitz, Ivo M. Vellekoop, Sid Assaworrorarit & Changhui Yang in Scientific Reports volume 3, Article number: 3543 (2013). The authors need to compare their work to that demonstrated in the mentioned paper. Additionally, secure communications enabled by key generation through the use of a pair of chaotic silicon photonic devices with ultrafast optical pulses using the approach mentioned in the reference above has been demonstrated before in a paper also not referenced (Brian C. Grubel, Bryan T. Bosworth, Michael R. Kossey, A. Brinton Cooper, Mark A. Foster, and Amy C. Foster, "Secure communications using nonlinear silicon photonic keys," Opt. Express 26, 4710-4722 (2018)). This work is the most similar to the manuscript (both utilize chaotic silicon photonic devices with ultrafast laser pulses, etc.), and the authors need to do a thorough comparison to put their work in context of this existing demonstration.

Additionally there are many practical concerns with the implementation of the system in its current form. As previously mentioned, although many details are not easily found, it appears that the silicon photonic devices require mechanical movement to modify initial conditions (specifically the angle of input coupling). Mechanical movements are extremely slow when compared to the repetition rate of the laser, so it is likely that many pulses will have identical initial conditions, even when the users intend to change rapidly. Another practical consideration is the requirement that both users have nearly identical optical sources (sources with identical spectra). This will prove to be a major challenge in any practical system. Finally, another major concern is the impracticality of providing irreversible behavior with the drying of droplets of unpurified water onto the silicon photonic device. This is not a practical way to imagine utilizing the system in a commercial communication application.

Reviewer #2 (Remarks to the Author):

The paper presents a method to achieve perfect secrecy cryptography using a Vernam cipher in a manner that: (i) cryptographic keys are generated from the light states that are exchanged via classical optical networks for a scalable, fast and economic implementation, (ii) these light states are generated by a chaotic random source, which is implemented by using inexpensive CMOS compatible silicon chips, and (iii) eavesdroppers who intercept parts of the light states cannot recover messages with uncertainty lower than 0.99 bit per transmitted bit. The paper is very well-written and, to the best of this reviewer's knowledge, justifies from every technical aspect that the presented method is a viable implementation that approaches the secrecy capacity to within 0.01 bit per transmitted bit. It is recommended to accept the paper for publication after addressing some minor issues as given below:

- As the paper aims at high throughput cryptographic key generation for economic significance, the rate of key generation per each transmission of the light state should accurately be evaluated. From this reviewer's interpretation, the key generation throughput increases as new overlapping repeated light states shown in Fig. 1b are observed more frequently from the optical channel. Therefore, in the absence of eavesdroppers, one may even find a condition (e.g., the rate of change of launching

conditions by Alice and Bob) that maximizes the rate of observation of new overlapping repeated light states, thereby maximizing the key generation throughput. On the other hand, in the presence of eavesdroppers, the rate at which new overlapping repeated light states are observed can be substantially reduced, hence the key generation throughput is reduced as well (if the number of generated bits of a key from each of the new overlapping repeated light states is constant). This should be clarified in the paper. In particular, it is questionable if the quantification of the key generation throughput in the first paragraph of page 20 is correct, since the throughput is dependent neither on the rate of change of launching conditions by Alice and Bob, nor on the rate of intercepted light states.

- It should be clarified whether the spatial displacement ϵ in Fig. 3c and Fig. 3e is a two-dimensional vector or a scalar. In addition, in Fig. 3e, it is unclear if ϵ has a fixed orientation or an independent and random orientation for every scatter s_i ; if the latter, ϵ should be denoted as ϵ_i , explicitly stating that it has a random orientation.

- The "oplus" operator first appears in page 6, but later defined in page 10. The operator should be defined in page 6 when it first appears.

- In the last paragraph of page 16, the figure numbers (b) and (c) should be corrected to (c) and (d).

Reviewer #3 (Remarks to the Author):

In the manuscript titled "Perfect secrecy cryptography via correlated mixing of chaotic waves in irreversible time-varying silicon chips", A. Di Falco et al. demonstrate a perfect secrecy cryptography system by making use of classical optical channels. In particular, they provide a physical implementation of the one-time pad (OTP) that, in spite of being patented by Vernam almost one century ago, has never been adopted in the digital realm due to the lack of a practical and secure way for users to exchange the encrypting key. In the reviewed manuscript, the authors claim the first physical realization of OTP that is compatible with established optical communications infrastructure and guarantees unbreakable security. For such a realization, A. Di Falco et al. use a protocol that exploits the transmission of correlated chaotic wavepackets, which are mixed in CMOS-compatible silicon chips, irreversibly modified in time, either before or after the communication, in order to ensure close to 100% security of the key. The chaotic property of the wavepackets is guaranteed by the fact that the generation uses a series of point scatterers microring resonators obtained by the (biometric) users' fingerprints. The authors demonstrate both theoretically and experimentally that, each time the chips are changed, the encrypting keys produced cannot be recreated again, not even by the same user. This is mainly due to the exploited scattering phenomena occurring within the fingerprint-based chips. Finally, the authors test the robustness of their protocol against any attempt of attack from an "eavesdropping", validating its security through thermodynamic principles, as well as through evaluating the entropy of the system.

The manuscript is very clear, with an accessible, yet scientific, language. Furthermore, the authors present in a very well detailed way the problems that affect secure message transmission, as well as the attempts to address them. Throughout the manuscript, A. Di Falco et al. provide a valid and strong argument to corroborate the robustness of their protocols from both an analytical/numerical and experimental point of view. They further discuss all potential attacks that the protocol can be subjected to, always proving a security of almost 100%.

A. Di Falco et al. describe a protocol that is the classic version of the quantum key distribution (QKD) which is based on the BB84 protocol (invented by Bennet and Brassard in 1984 and that makes, in this work, use of photons). Indeed, A. Di Falco et al. show two communicating parts, i.e. Alice and Bob, as well as Eve, who try to intercept the message. As mentioned by the authors, a development of the quantum protocol BB84, despite guaranteeing much more security respect with their classical counterparts, is currently limited by, e.g. the impossibility of amplifying photons, and so forth. Moreover, quantum-based communication protocols are significantly more expensive than their classical counterparts. With the results presented in the reviewed manuscript, the authors definitely pave the way towards the possibility of transmitting encrypted data and messages in a well-established and fiber-based classical communication network, while still preserving key security.

The reviewed work is novel and can have a significant impact not only on the scientific community, but also in terms of practical implementation for secure telecommunication over the existing infrastructure in a relatively immediate future.

For these reasons, the manuscript "Perfect secrecy cryptography via correlated mixing of chaotic waves in irreversible time-varying silicon chips" by A. Di Falco et al. addresses the criteria for being published in Nature Communications.

However, there are a few points that need to be addressed. As I mentioned before, the authors present the classic version of a QKD protocol only in the introduction, where they quickly describe the limits of quantum protocols. As a matter of fact, quantum protocols guarantee for more security than their classical counterparts due to the quantum mechanical laws (e.g. no cloning theorem, measurements that causes a collapse of the quantum states, thus allowing Alice and Bob to know whether their message has been intercepted or not). Furthermore, the 'quantum revolution' is making several steps towards the development of quantum-based technology. Therefore, at least in the description of the exchanged key between Alice and Bob, and in the evaluation of their 'outcomes', it is worth doing a parallel comparison with the quantum experiment. Of course, the system that the authors use cannot be adapted to the quantum regime. However, a parallel comparison between classical and quantum could make the manuscript also interesting and helpful for QKD protocols and thus for the quantum scientific community at large.

The authors discuss the possibility of deterministic errors that might arise in the bitstream sequence between Alice and Bob due to active attacks. In this case, there is not information acquired by Eve during the attack resulting in a very secure cryptography system. Those errors are small and scale as the number of chaotic states available in Alice and Bob fingerprint devices. However, those need to be properly corrected in order to preserve the performance of the key distributions, the security, and the length of the cryptographic key. In this regard, the authors mention that techniques of information reconciliation and privacy amplification are available and can be integrated in their system, due to the public nature of the communication channel. The question that might arise, especially from those who are not familiar with error detection/correction techniques, is concerning whether or not these techniques are some sort of 'weakness' for the system, allowing potential security breach revealing the key. This point should be clarified by the authors.

General comments:

- The average Lyapunov exponent is labelled with the Greek letter ' μ '. In the text there is an

occurrence with 'Mu' being replaced by 'h'.

- It is strongly suggested to check all the figures captions/legends for misprints.

Author Response to Reviews of

Perfect secrecy cryptography via correlated mixing of chaotic waves in irreversible time-varying silicon chips

RC: Reviewer Comment, AR: Author Response, □ Manuscript text, “ ” Reference quotation

Reviewer #1

RC: In the manuscript “Perfect secrecy cryptography via correlated mixing of chaotic waves in irreversible time-varying silicon chips,” the authors demonstrate key distribution that provides perfect secrecy on a classical communication channel. They utilize chaotic silicon integrated photonic chips to act as a unique, random, and irreversibly modifiable filters for the ultrafast optical pulse. The filter behavior, which is dynamically changed in time by the user, serves as the source of the shared key between the two parties. This shared key can then be used to encrypt a message using the one-time pad (OTP) approach. Notably, in contrast to quantum key distribution (QKD), the authors’ approach utilizes a classical signal.

The main novelty of the manuscript is the ability to distribute a shared key for perfect secrecy communications using classical signals. This is a very exciting possibility: to be able to do key distribution using classical means and provide perfect secrecy has not been accomplished before. This is extremely exciting compared to the only proven alternative of a quantum approach. However, the manuscript is extremely confusing and needs a great deal of simplification in order to provide the reader with a clear picture of the technique and how it is able to achieve this perfect secrecy.

The silicon integrated photonic devices utilized as the source of a chaotic filter are incremental, and are a forced recreation of the free-space scattering optical PUFs previously demonstrated in the literature, and therefore suffer from many of the same shortcomings as their free-space counterparts; chief among them being the need for free-space alignment due to their multimode nature. The research presented in the manuscript is novel and interesting, however it is not suitable for publication at this time in its current form. The manuscript is bogged down with extraneous details, and the crux of the functionality of the system is hidden in the details of the supplementary files. I had to read the paper several times before I could understand how the system works. After the reader finally is able to understand how the key is generated, the next most obvious train of thought is to question why the system is secret. The most natural way to attack the system is to have an eavesdropper passively measure the transmissions and attempt to recreate the key. This most basic attack is skipped over in the manuscript, and after the authors address many ancillary attack approaches, they finally direct the reader to the supplementary files to try to understand the crux of how the key creation can remain secret. This will be addressed in more detail below.

The most obvious way to attack the proposed system would be for an eavesdropper to passively eavesdrop and attempt to measure the signal. The shared key is generated through the combined response of two chaotic chips, and due to reciprocity, the combined response is the same regardless of what direction, therefore both users, Alice & Bob, can measure the same response from light originating from the opposite user’s location. The essential problem to provide security is ensuring that an eavesdropper (Eve) cannot also measure that combined response. In this case, the reason is because the key extraction is in essence made very sensitive to

the SNR of the measurement, and each of the users have a better SNR than the eavesdropper. This means that the transmission must be performed with the proper amount of injected noise such that the users can extract the key without error, whereas no information about the key is leaked to Eve. This (my last two sentences) is the crux of the manuscript, and it is buried in the details of the supplementary file. It is completely absent from the figures of the manuscript, and should be included in the overall system figure (figure 1).

AR: *We acted on the Referee's comments and improved the manuscript by restructuring its content following the Referee's guidelines. The summary of the changes is as follows:*

1. *We streamlined the narrative and divided the manuscript's security analysis into different sections:*

- *Perfect secrecy of the cipher*
- *Perfect secrecy of the key distribution scheme*
- *Security against time domain attacks*
- *Security against spectral attacks*

with each section focusing on a different type of attack. Following the Referee's request, we moved the discussion on spectral attacks (with the included analysis on the SNR) at the beginning of the paper and in the new section "Security against spectral attacks" spanning lines 231–298.

2. *We discussed spectral attacks in a figure at the beginning of the manuscript, as the Referee suggested. The new figure, revised Fig. 2 (reported below as Fig. A1), illustrates results on the system's security for both time-domain and spectral attacks, facilitating the reader's access to this information and thus simplifying the discussion.*

The extensive revision discussed in points 1-2 clarifies, in a single section of the manuscript, the security of the scheme against any possible attack and substantiates the limit of perfect secrecy tackled by the work.

3. *Concerning the Referee's question on the free-space coupling, we revised the text and clarified that the structure proposed is integrated on-chip and, as such, it does not suffer from any of the limitations of traditional free space systems. Integrated light coupling is indeed possible by using multitapered channel couplers, a mature technology developed in the silicon photonics community, as detailed in the experimental work of:*

- (a) *Doany, F. E. et al., Multichannel high-bandwidth coupling of ultradense silicon photonic waveguide array to standard-pitch fiber array. Journal of Lightwave Technology 29, 475–482 (2010).*
- (b) *Almeida, V. R., Panepucci, R. R. & Lipson, M. Nanotaper for compact mode conversion. Optics letters 28, 1302–1304 (2003).*

and summarized in the review work of Vlasov:

- (c) *Vlasov, Y., Green, W. M. & Xia, F. High-throughput silicon nanophotonic wavelength-insensitive switch for on-chip optical networks. Nature photonics 2, 242 (2008).*

To clarify this point, we added a new section titled "Integrated on-chip implementation with ultrafast light modulation" in the supplementary information (Sec. VII). The section discusses a possible implementation of an integrated device with on-chip coupling from end to end and ultrafast modulation of the input signal, addressing also the second-last question of the Referee. The section reads as follow:

Figure A1: Revised Fig. 2 of the main text. **Protocol security against time-domain and spectral attacks.** (a) Uncertainty per bit measured by an ideal attacker for all possible types of attempted time-domain key reconstruction: \otimes (red line), $+$ (green line), \cdot (blue line), when the users mix random wavepackets with increasing number M of different frequencies. (b-c) Statistics of Intensity of single wavepackets for (b) aperiodic and (c) chaotic cases. (d-f) Results of spectral attacks: Eve's uncertainty per bit (solid green line), BER between Alice and Bob keys (solid red line) for different standard deviations σ_{AB} between the power density spectra of the combined states measured by Alice and Bob, as a function of the number of bit N extracted for each spectral point measured. The figure reports average values (solid lines) and standard deviations (error bars).

Figure A2: Supplementary Fig. 6. Block scheme for integrated ultrafast light modulation with on-chip light couplings from end-to-end of the communication line: (EO: electro-optic modulator, SM: switch matrix).

The integrated nature of the fingerprint chips enables ultrafast modulations with integrated light coupling. While the implementation of a high-dense device of this type goes beyond the scope of this paper, we here discuss a possible realization with the technology currently available.

The results of Fig. 4c show that the fingerprint chip, at every spatial point, generates uncorrelated data sequences from TE and TM polarized light. The use of different input position and light polarizations can be combined into the block scheme of Supplementary Fig. 6. The initial stage is composed of a modified Mach-Zender, excited by an unpolarized source, or equivalently a combination of TE and TM polarized light sources. The Mach-Zender interferometer is composed by a y-junction fiber polarization splitter, followed by two ultrafast electro-optical (EO) modulators and a final y-junction for recombining the two polarization arms. The output fiber is then connected to a $1 \times N$ optical switch matrix [5].

Both EO modulators and switch matrices can provide modulations up to hundreds of GHz and with hundreds of channels with the technology currently available [5], thus providing ultrafast selection of the input position on the fingerprint chip without the use of any mechanical component. The output light from the optical switch is then directly fed on to the fingerprint chip via multi-tapered channels couplers [6-8], providing on-chip couplings from end-to-end to the communication line. With this integrated structure, it is possible to achieve the generation of $4 \cdot N^2$ different spectra at each communication.

The new section contains Refs. (a)-(c) as Refs. [6-8], a new Ref. [5]:

- *Stabile, R., Albores-Mejia, A., Rohit, A. & Williams, K. A. Integrated optical switch matrices for packet data networks. Nature Microsystems & Nanoengineering 2, 15042 EP – (2016)*

and a new figure, Supp. Fig. 6 (here reported as Fig. A2), illustrating the integrated scheme.

The section is referenced on line 407 of the main text:

A fully integrated structure with on-chip coupling from end to end is discussed in Supplementary Sec. VII.

RC: Many aspects of the system are very similar to previously published work and the authors do not put their work in the context of the existing literature (nor are the two very relevant works referenced). Secure communications on classical optical channels have been demonstrated before, with the system in “Physical key-protected one-time pad,” by Roarke Horstmeyer, Benjamin Judkewitz, Ivo M. Vellekoop, Sid Assaworranit & Changhui Yang in Scientific Reports volume 3, Article number: 3543 (2013). The authors need to compare their work to that demonstrated in the mentioned paper. Additionally, secure communications enabled by key

generation through the use of a pair of chaotic silicon photonic devices with ultrafast optical pulses using the approach mentioned in the reference above has been demonstrated before in a paper also not referenced (Brian C. Grubel, Bryan T. Bosworth, Michael R. Kossey, A. Brinton Cooper, Mark A. Foster, and Amy C. Foster, "Secure communications using nonlinear silicon photonic keys," *Opt. Express* 26, 4710-4722 (2018)). This work is the most similar to the manuscript (both utilize chaotic silicon photonic devices with ultrafast laser pulses, etc.), and the authors need to do a thorough comparison to put their work in context of this existing demonstration.

AR: *We followed the Referee's suggestion and revised the paper by providing a thorough analysis of the novelty of this work in comparison with existing literature on PUFs. The main points are summarized below:*

1. *From a general perspective, all existing literature on PUFs, including the 2 articles mentioned by the Referee:*

(a) *"Physical key-protected one-time pad," by Roarke Horstmeyer, Benjamin Judkewitz, Ivo M. Vellekoop, Sid Assaworrorarit & Changhui Yang in Scientific Reports volume 3, Article number: 3543 (2013),*

(b) *Brian C. Grubel, Bryan T. Bosworth, Michael R. Kossey, A. Brinton Cooper, Mark A. Foster, and Amy C. Foster, "Secure communications using nonlinear silicon photonic keys," Opt. Express 26, 4710-4722 (2018),*

do not tackle the limit of perfect secrecy and do not offer unconditional security, while mainly focusing on providing advantages over electronic systems in terms of memory capacity, due to the use of volumetric physical data storage vs electronic databases.

The security of all PUF systems (including Refs. (a) and (b), cited as Ref. [40-41]) is based on two arguments: the condition that the random object used in the scheme is kept secret, and the conjecture that the scattering object is unfeasible to clone. These arguments do not provide perfect secrecy, but mathematical or probable security. The following recent work:

(c) *Helfmeier, C., Boit, C., Nedospasov, D. & Seifert, J.-P. Cloning physically unclonable functions. In Hardware-Oriented Security and Trust (HOST), 2013 IEEE International Symposium on, 1-6 (IEEE, 2013),*

showed that it is indeed experimentally possible, with the technology currently available, to clone a physical unclonable function. This point is also emphasized in the introduction of Ref. [36] [Quantum-secure authentication of a physical unclonable key. Optica 1, 421-424 (2014)]:

"Physical unclonable function (PUF) is a physical object that cannot feasibly be copied because its manufacture inherently contains a large number of uncontrollable degrees of freedom. Making a sufficiently accurate clone or concocting a device that mimics its physical behavior is infeasible, though not theoretically impossible, given the properties of PUFs"

and makes no confusion on the conjectural nature of this argument, depending only to the availability of technological resources.

The second assumption, keeping the PUF secret to an adversary, does not hold in the case of perfect secrecy. Perfect secrecy implies discussing the system security with the Kerckhoffs' principle: in the case where the enemy knows all details of the system, accesses the system before or after the communication copying all components (regardless how technologically difficult and lengthy it could be), controls the communication channels and accesses all information exchanged by the users on

public channels.

The lack of perfect secrecy is well acknowledged in the papers highlighted by the Referee. Reference [40] precisely acknowledges that the system provides mathematical security:

“Unlike any other secure communication setup our C-PUF protocol can meet these strict requirements, with a practical security limit set only by the amount of time it takes to mathematically characterize the highly complex structure of its volumetric scatterer”

while Ref. [41] clearly specifies that, in addition to the hard requirement that the PUF structure is kept secret to the adversary, the security provided is probable and based on the argument that the sample cannot be copied via conventional e-beam lithography:

“ To investigate this approach, we fabricated six unique silicon photonic micro-cavity PUF designs (variations on a 30-micron diameter silicon disk with a chamfer) and two exact copies (clones) of each PUF using electron beam lithography and standard nanofabrication techniques...In addition, two copies of every cavity are fabricated on the same SOI die and in the same fabrication run, permitting analysis of PUF clonability...”

Due to these limitations all current PUF schemes —as their electronic counterparts— do suffer from all security concerns arising from the lack of perfect secrecy, opening to a series of possible attacks, e.g., digital attacks:

“ When they are read out classically, PUFs are vulnerable to a class of attacks that we will refer to as digital emulation [Fig. 1(b)]. Here the adversary has knowledge of the key’s properties either from physical inspection of the key or by access to the challenge–response database. He intercepts challenges and is able to provide the correct responses by looking them up in his database. This is a highly relevant scenario as accessible databases are notoriously difficult to protect. So far the only defense against digital emulation is to deploy various sensors that try to detect if some form of spoofing is going on. This leads to an expensive arms race in which it is difficult to ascertain the level of security. (Quoted from Ref. [36])”

or more dangerous post-data-processing:

“Consequently, conventional cryptography is vulnerable to unanticipated advances in hardware and algorithms, as well as to quantum code breaking, such as Shor’s efficient algorithm for factoring. This is potentially problematic as government and trade secrets are kept for decades. An eavesdropper, Eve, may simply save communications sent in 2014 and wait for technological advances. If she is able to factorize large integers [clone PUF structures,...] in say 2100, she could retroactively break the security of data sent in 2014.

In contrast, quantum key distribution (QKD), the best-known application of quantum cryptography, promises to achieve the Holy Grail of cryptography — unconditional security in communication. In unconditional security, or more precisely ϵ -security (see the section “Security model of QKD”), Eve is not restricted by computational assumptions, but only by the laws of physics.”

Quoted from Ref. [7] (Secure quantum key distribution, Nature Photonics volume 8, pages 595–604 (2014)).

Currently, only QKD can prevent these types of threats and can guarantee a theoretically unbreakable

scheme. In this manuscript, conversely and for the first time, we tackle the limit of perfect secrecy with a fully classical structure, attaining the same security previously possible only by QKD, while retaining all the advantages of classical communications (speed, cost & scalability). This is an element of considerable novelty compared to existing work on classical PUF.

To clarify this point in the manuscript, we revised lines 86–97:

The use of static light scatterers in information security is introduced in [31] and offers computational and probable security in both authentication problems and cryptographic key generation, providing advantages over electronic schemes in terms of volumetric physical data storage vs standard electronic databases [32-41]. The security of schemes based on complex scattering structures relies on two conditions: i) the physical scattering object is kept secret to an adversary, ii) the assumption that this structure cannot be cloned. These arguments do not offer unconditional security and are subject to the same security concerns of electronic schemes. While recent work demonstrated that it is indeed experimentally possible to clone a physically unclonable function [42], perfect secrecy requires proving the system security in the limit where the adversary accesses the system before or after the communication, copying all the system's parts. In this work, as we are discussing the limit of perfect secrecy, we remove conditions i)-ii) and we consider an adversary with the technology to clone any type of scatterer.

and added article (c) as Ref. [42] in the revised text.

The achievement of the limit of perfect secrecy in the work proposed in this manuscript is accomplished by the introduction of many elements of technical novelty, making this system different in many relevant aspects with respect to current PUF:

- (a) **New communication setup:** a new scheme that does not require first encounter or initial setup.
- (b) **New scattering object:** the use of a time varying complex scatterer subjected to irreversible thermodynamic transformations.
- (c) **New communication protocol:** a new classical protocol that mimics the structure of the original BB84 quantum scheme developed for QKD.
- (d) **New key generation:** a new approach that is not based on dictionaries or electronic databases.

Below, we discuss these points one by one in detail.

- a. **New communication setup.** Both schemes presented in Ref. [40-41] require a first encounter or secure communication, e.g. performed with QKD (Ref. [40]). This initial setup phase is independent from the authentication of the users and has to be performed every time the scatterer is used for the first time. This condition is very impractical, especially in conjunction with costs and scalability issues of QKD, as well as the finite amount of keys that can be extracted from each PUF.

As discussed in the manuscript Section “Vernam cipher on classical channels: theory”, the communication algorithm developed in this manuscript does not require any first encounter or secure initial communication (apart from the one required for authenticating the users).

To better emphasize this point, we revised the text on lines 451–452:

Beyond the initial communication required for authenticating the users, the system does not require electronic databases, private keys or confidential communications.

and on lines 117–119:

At variance with classical schemes based on complex scatterers [40-41], the protocol here presented does not require first encounter or initial secure communication among the users (apart from authentication) thus providing a classical alternative to QKD.

This possibility is enabled by the use of a new communication scheme, as commented at point c).

- b. **New scattering object.** *The work here presented uses a time varying, irreversible complex scattering object, contrary to existing PUF schemes that uses static scatterers. An irreversible, time varying complex scatterer guarantees the continuous generation of unique keys and, and as discussed in the manuscript section “Perfect secrecy of the key distribution scheme”, it also guarantees that the attacker will never be able to copy the structure at the moment of the communication.*

None of the existing scheme can employ this approach: it would imply that the users meet personally at every communication for the required initial setup, or equivalently, use QKD at every communication, thus invalidating the need for an additional PUF based cryptographic protocol.

To clarify this point, we revised lines 156–171:

Another possibility is to make an identical copy of the system in all its parts, and to attempt the search at the next communication. This is a major vulnerability of all current classical cryptographic schemes. The system presented in this work, on the contrary and due to the use of irreversible thermodynamic transformations, is perfectly protected by such attack. The search task, in fact, requires Eve to generate the same chaotic scatterers as of Alice and Bob’s, so that their transformations are cloned prior to the communication. As the chips are in equilibrium with the environment (point 2), this task requires replicating the surroundings of Alice and Bob’s chips. This condition is essential for enabling Eve’s copied chips to reach the same equilibrium state of the original chips of Alice and Bob. The second law of thermodynamics makes this operation not physically possible. Eve, in fact, cannot replicate the exact time at which Alice and Bob perform their transformations. If Eve does the transformation after Alice or Bob, the environment will be different, as it existed at least one irreversible transformation in time (the one of Alice or Bob) that increased its entropy; and vice-versa if Eve performs the transformation before the users. It will be therefore impossible for Eve to clone the transformation of Alice and Bob. Due to points 3-4, Eve will generate new chaotic scatters that are exponentially different from the ones that Alice or Bob are using and, as such, useless.

- c. **New communication protocol.** *The protocol derived in this work (as also observed by Referee 3), is a classical version of the original BB84 scheme of QKD based on the transmission of single photons states with random polarizations, here mimicked by random scattering events of a large ensemble of photons in the classical limit. In this sense, the correlations between Alice and Bob’s quantum mechanical states take place by the reciprocity theorem of electromagnetic waves, propagating in the bidirectional channel created by Alice and Bob. This allows to overcome many limitations of current classical schemes (such, as, e.g. the first encounter commented at point a) that are not present in QKD protocols. To clarify the manuscript on this point, we added on lines 113–116:*

The communication protocol described in Fig. 1 can be regarded as a classical version of the original BB84 QKD scheme developed by Bennett and Brassard [45], in which the scatterers act as generator of random states, and the reciprocal communication line provides correlated measured states to the users Alice and Bob.

and added Ref. [45]:

- Bennett, C. H. & Brassard, G. *Quantum cryptography: Public key distribution and coin tossing.* In *Proceedings of IEEE International Conference on Computers, Systems, and Signal Processing*, 175 (1984)

d. **New key generation.** *The work [41] requires the creation of a public dictionary, and relies on a lengthy process of challenge/response with the dictionary in order to generate a key. This implies that the time required for key sharing is longer than the time to exchange the message. It also opens to possible attacks if the database is exposed to an attacker with some knowledge of the PUF. The generation of dictionaries, commonly employed in PUFs, are considered a security limitation due to their digital nature (please see the earlier text quoted from Ref. [36] on digital emulation).*

The system proposed in this manuscript, conversely, does not require databases and exploits a fully physical interaction to generate the key. As discussed in the manuscript section “Key generation throughput”, the scheme can distribute a One-time Password (OTP) key of the same bit length of the message from the transmission of around 1/1000 of the bit data contained in the original message. This is another element of considerable novelty, also with respect to QKD, in which the communication speed is one of the bottleneck of the system. To better emphasize these points in the text, we revised lines 21–24:

The keys generated with this protocol require the transmission of an amount of data that, in the absence of active eavesdropping, can be as small as $\approx 1/10^3$ of the message’s length, and are generated at the distal end of each user without never being visible in the communication line.

lines 54–58:

Due to the impossibility of amplifying single photons [20], quantum networks are currently unable to scale up globally; their data transfer is considerably slower than classical optical communications, which already count with hundreds of high-bandwidth intercontinental lines, communication speed close to the light limit and massive investments for the next years [21-26].

and line 451:

...the system does not require electronic databases, private keys or confidential communications.

RC: Additionally there are many practical concerns with the implementation of the system in its current form. As previously mentioned, although many details are not easily found, it appears that the silicon photonic devices require mechanical movement to modify initial conditions (specifically the angle of input coupling). Mechanical movements are extremely slow when compared to the repetition rate of the laser, so it is likely that many pulses will have identical initial conditions, even when the users intend to change rapidly.

AR: We acted on the Referee's comment and provided a revision of the text to address the Referee's question. The summary of changes is as follows:

1. We clarified in the revised text that the change in the input condition in the chips is not restricted to position, but it is implemented independently at each user's side by any physical mean, including e.g., position, angle, polarization, time modulation, etc... As rigorously demonstrated both theoretically and experimentally in the "Integrated biometric Silicon chips" Section and in Supplementary Sec. V-VI, in fact, any variation in the input condition of light injected in the scatterers leads to uncorrelated output dynamics.

To clarify this point in the text, we revised lines 70–73:

These states are generated from the chaotic scattering of broadband pulses with different frequencies $\omega_1, \dots, \omega_m$, launched inside each chip with diverse input conditions n and n' (position, angle, polarization, time modulation,...) arbitrarily chosen by Alice and Bob.

2. To further substantiate this point theoretically, we revised Fig. 4 of the main text (reported here as Fig. A3) and analyzed the behavior of the structure for impinging light with different TE and TM polarizations at each spatial input position. We then calculated the entropy correlation matrix between TE-TE, TE-TM, TM-TM sequences and presented the results in revised Fig. 4c. The results show that, for every input position, sequences generated from TE and TM polarized light are completely uncorrelated, in agreement with the theoretical analysis. To clarify this point in the manuscript, we revised the text on lines 352–360:

We calculated transmitted electromagnetic spectra for both TE and TM polarized point sources of 150 fs duration, centered at the wavelength $\lambda = 1550$ nm, and launched at $x = y = 0$ with displacements y_1, y_2, \dots along y within 1 μm range with 20 nm resolution. For each input position, we computed the transmitted energy spectrum and transformed it into a binary sequence by the AHB technique with the parameters of Supplementary Sec. I. We then computed the entropy correlation matrix \mathcal{H} , with elements $\mathcal{H}_{ij} = -d_{ij} \log_2 d_{ij} - (1 - d_{ij}) \log_2 (1 - d_{ij})$ being the Shannon information entropy of the hamming distance d_{ij} among the bit sequences i and j arising from PDS obtained by input shifts y_i and y_j . The entropy correlation matrix is strongly diagonal (Fig. 4c), showing that the generated bit sequences are completely uncorrelated.

3. We added a new supplementary section VII titled "Integrated on-chip realization with ultrafast light modulation". The section, discussed in the answer of the first point in the Referee's report, clarifies that, with the available optical technology, the structure experimentally demonstrated in the work can be scaled up to ultrafast GHz light modulation.

RC: Another practical consideration is the requirement that both users have nearly identical optical sources (sources with identical spectra). This will prove to be a major challenge in any practical system.

AR: We clarified in the revised text that the users' sources are not required to be identical; on the contrary, their differences are here used as a primary source for uncertainty, as discussed in the revised section "Security against spectral attacks".

To clarify these points in the revised text, we added on lines 73–75:

Figure A3: Revised Fig. 4 of the main text. **Wave analysis of fingerprint chips from FDTD simulations.** (a) Electromagnetic energy distribution in a resonator made by air pillars on a silicon substrate for an input pulse at $\lambda = 1550$ nm and 150 fs long. (b) Probability distribution of the electromagnetic energy inside the resonator. (c) Correlation entropy between the binary sequences generated from transmitted electromagnetic spectra obtained by shifting the launching position of the input beam within $1 \mu\text{m}$ from $x = y = 0$ with displacements of 20 nm for TE and TM polarized excitation. (d) Average correlation entropy for displacements within ± 150 nm.

Alice and Bob's pulses are not required to be identical: their differences constitute the main source of uncertainty and set the desired communication bit error rate (BER) in the communication.

and on lines 245–248:

In an experimental implementation, the main statistical components of uncertainty is the fluctuation of the source Δ_S (see Supp. Section III). This is also a general case the users can always choose, as the source is controlled by the users at the time of the communication. In the limit of acceptable bit error rate (BER), Δ_S can always be increased if necessary by adding artificial time fluctuations.

while deferring technical calculations to supp. section III:

The uncertainty contained in the reciprocal power density spectra of Alice and Bob arises from the following uncorrelated components:

- *Statistical fluctuations Δ_S of source spectrum.*
- *Noise thermal fluctuation Δ_C of the communication channel.*
- *Fluctuations Δ_A, Δ_B in the input conditions.*
- *Uncertainty Δ_M of the measurement apparatus.*
- *The coupling coefficient $\alpha(\omega)$: this is a systematic source of uncertainty (type B of NIST guidelines [1]). To evaluate $\alpha(\omega)$, the user must measure the absolute value of a spectrum and normalize it. To do so, the user must use another measurement instrument, which introduces an unknown coupling coefficient. This argument can be iterated at infinity, without the user knowing the first coefficient. It is the same problem of measuring the exact temperature of a body: one needs to know in advance the temperature of the thermometer, as it reads the equilibrium temperature between the coupled environments of the thermometer and the body.*

Typical optical sources have statistical fluctuations in power amplitudes of the order of few percents. Just to make a specific example, the CL band tunable source T100S-HP from EXFO ensures an absolute wavelength accuracy better than ± 20 pm and a power repeatability sweep-to-sweep better than ± 0.05 dB, which implies differences as small as $\pm 1\%$ at every wavelength. The optics/optomechanics available today are stable to fluctuations below 10 nm, providing very little variations in measured optical observable if compared to source fluctuations. Thermal fluctuations in the communication channel are also negligible and typically many orders of magnitude smaller than the source's amplitude. Measurement instruments available today have a dynamic range in the range of tens of dB and a spectral accuracy in the range of tens of picometers, thus providing very precise reading with standard deviation much smaller than sources amplitude variations. In an typical experimental realization, we are therefore in the situation where $\Delta_{A,B,C,M} \ll \Delta_S$.

RC: Finally, another major concern is the impracticality of providing irreversible behavior with the drying of droplets of unpurified water onto the silicon photonic device. This is not a practical way to imagine utilizing the system in a commercial communication application.

AR: We acted on the Referee's comment and revised lines 377–379:

In a commercial oriented application, this step can be accomplished by the deposition of solid state scatterers, such as, e.g. doped hydrogels, which are mechanically deformable with temperature, pressure, light or electrical signals [63-65].

and added Refs [63-65]:

- *Shin, D. et al. Scalable variable-index elasto-optic metamaterials for macroscopic optical components and devices. Nature communications 8 , 16090 (2017).*
- *Ionov, L. Hydrogel-based actuators: possibilities and limitations. Materials Today 17, 494–503 (2014).*
- *Walker, E. L., Wang, Z. & Neogi, A. Radio-frequency actuated polymer-based phononic meta-materials for control of ultrasonic waves. NPG Asia Materials 9, e350 (2017).*

Reviewer 2

RC: The paper presents a method to achieve perfect secrecy cryptography using a Vernam cipher in a manner that: (i) cryptographic keys are generated from the light states that are exchanged via classical optical networks for a scalable, fast and economic implementation, (ii) these light states are generated by a chaotic random source, which is implemented by using inexpensive CMOS compatible silicon chips, and (iii) eavesdroppers who intercept parts of the light states cannot recover messages with uncertainty lower than 0.99 bit per transmitted bit. The paper is very well-written and, to the best of this reviewer's knowledge, justifies from every technical aspect that the presented method is a viable implementation that approaches the secrecy capacity to within 0.01 bit per transmitted bit. It is recommended to accept the paper for publication after addressing some minor issues as given below:

AR: *We thank very much the Referee for the positive assessment of the work and for the list of constructive comments, which helped to improve the quality of the manuscript.*

RC: As the paper aims at high throughput cryptographic key generation for economic significance, the rate of key generation per each transmission of the light state should accurately be evaluated. From this reviewer's interpretation, the key generation throughput increases as new overlapping repeated light states shown in Fig. 1b are observed more frequently from the optical channel. Therefore, in the absence of eavesdroppers, one may even find a condition (e.g., the rate of change of launching conditions by Alice and Bob) that maximizes the rate of observation of new overlapping repeated light states, thereby maximizing the key generation throughput. On the other hand, in the presence of eavesdroppers, the rate at which new overlapping repeated light states are observed can be substantially reduced, hence the key generation throughput is reduced as well (if the number of generated bits of a key from each of the new overlapping repeated light states is constant). This should be clarified in the paper. In particular, it is questionable if the quantification of the key generation throughput in the first paragraph of page 20 is correct, since the throughput is dependent neither on the rate of change of launching conditions by Alice and Bob, nor on the rate of intercepted light states.

AR: *We followed the Referee's suggestion and added a new section in the main text titled "Key generation throughput", which discusses this point in detail, calculating also the condition of maximal throughput as suggested by the Referee. The new section reads as follows:*

In the absence of active eavesdropping, the number of pulses N_p to be transmitted for generating an OTP key is:

$$N_p = \frac{L_m}{N_b \cdot P^{(ov)}}, \quad (1)$$

and equal to the ratio between the length of message L_m and the number of bits N_b extracted from each combined state $A_n \oplus B_{n'}$, multiplied by the probability $P^{(ov)}$ of observing a repeated state at the same time position in Alice and Bob's sequences. The maximum key generation throughput is observed when $P^{(ov)}$ is maximal.

Supplementary Fig. 4 shows the probability $P^{(ov)}$ for different values of the users probabilities $P^{(Alice)}$ and $P^{(Bob)}$ to change state after each step. A maximum value of $P^{(ov)} = 10\%$ is obtained when $P^{(Alice)} = P^{(Bob)} = 0.25$. In this condition, by using standard detectors at 1024 bits and by extracting 10 bits per spectral sample, the protocol requires the transmission of $N_p = \frac{L_m}{1024}$ pulses, corresponding to $\approx 1/1000$ of the length of the message.

In the presence of active eavesdropping, the key generation throughput decreases due to the additional error states that the eavesdropper is introducing. According to the results of Supplementary Section II and Supplementary Fig. 1, the throughput variation introduced by these attacks is in the range of a few percents.

and included a new figure, Supplementary Fig. 4 (here reported as Fig. A4). We then clarified this point in the Abstract, as requested:

The keys generated with this protocol require the transmission of an amount of data that, in the absence of active eavesdropping, can be as small as $\approx 1/10^3$ of the message's length, and are generated at the distal end of each user without never being visible in the communication line.

RC: It should be clarified whether the spatial displacement ϵ in Fig. 3c and Fig. 3e is a two-dimensional vector or a scalar. In addition, in Fig. 3e, it is unclear if ϵ has a fixed orientation or an independent and random orientation for every scatter s_i ; if the latter, ϵ should be denoted as ϵ_i , explicitly stating that it has a random orientation.

AR: We acted on the Referee's remark and amended the paper on lines 326–329:

Figure 3c shows the propagation of light rays of two input conditions $(\mathbf{x}_0, \mathbf{n})$ (solid red line) and $(\mathbf{x}_0 + \epsilon, \mathbf{n})$ (solid green line), having the same initial orientation \mathbf{n} and spatially displaced by a random vector ϵ with $|\epsilon| = \epsilon_{min} = 2.2 \cdot 10^{-16}$ the smallest floating point number representable at the computer.

and lines 338–341:

The plot shows the dynamics of three identical input conditions (solid blue, red, and green lines) launched in three different fingerprint resonators implemented by randomly shifting the positions of each scatterer s_i by a vector with random orientation ϵ_i , with $|\epsilon_i| = \epsilon_{min}$.

We also amended Fig. 2e (reported here as Fig. A5).

Figure A4: Supplementary Fig. 4. Probability $P^{(ov)}$ of observing repeated states at the same position in Alice and Bob's sequences as a function of the users probabilities $P^{(Alice)}$ and $P^{(Bob)}$ to change state after each repeated communication. The calculations are performed on a statistical sample of 1000 sequences, each of 100000 states.

Figure A5: Revised Fig. 2 of the main text. **Integrated fingerprint silicon chips design and chaotic analysis.** (a-b) Biometric fingerprint acquisition and transformation into a chaotic resonator (c). (c) Dynamics of two trajectories displaced by the smallest number representable at the computer. (d) Averaged Lyapunov exponent $\langle \mu \rangle$ in the phase space of possible input conditions (blue line) and percent of the phase space filled with chaotic dynamics for different radius r of the scatterers (green line). (e) Dynamics of three identical input conditions in three infinitesimally different resonators.

RC: The "oplus" operator first appears in page 6, but later defined in page 10. The operator should be defined in page 6 when it first appears.

AR: *We followed the Referee's suggestion and revised the manuscript on lines 103–105:*

For instance, when Alice sends a chaotic wavepacket A_n to Bob, he measures an optical observable, e.g. the intensity $|A_n \oplus B_{n'}|^2$ associated with the combined light state $A_n \oplus B_{n'}$ (\oplus is the operator that combines the states after the propagation over the channel)

RC: In the last paragraph of page 16, the figure numbers (b) and (c) should be corrected to (c) and (d).

AR: *We fixed the typo as requested:*

Figures 6c-d analyzes the transmission for one set of input conditions showing: (c) the power density spectra sent by Alice and Bob and (d) the spectra measured at the distal end by the users.

Reviewer 3

RC: In the manuscript titled "Perfect secrecy cryptography via correlated mixing of chaotic waves in irreversible time-varying silicon chips", A. Di Falco et al. demonstrate a perfect secrecy cryptography system by making use of classical optical channels. In particular, they provide a physical implementation of the one-time pad (OTP) that, in spite of being patented by Vernam almost one century ago, has never been adopted in the digital realm due to the lack of a practical and secure way for users to exchange the encrypting key. In the reviewed manuscript, the authors claim the first physical realization of OTP that is compatible with established optical communications infrastructure and guarantees unbreakable security. For such a realization, A. Di Falco et al. use a protocol that exploits the transmission of correlated chaotic wavepackets, which are mixed in CMOS-compatible silicon chips, irreversibly modified in time, either before or after the communication, in order to ensure close to 100% security of the key. The chaotic property of the wavepackets is guaranteed by the fact that the generation uses a series of point scatterers microring resonators obtained by the (biometric) users' fingerprints. The authors demonstrate both theoretically and experimentally that, each time the chips are changed, the encrypting keys produced cannot be recreated again, not even by the same user. This is mainly due to the exploited scattering phenomena occurring within the fingerprint-based chips. Finally, the authors test the robustness of their protocol against any attempt of attack from an "eavesdropping", validating its security through thermodynamic principles, as well as through evaluating the entropy of the system.

The manuscript is very clear, with an accessible, yet scientific, language. Furthermore, the authors present in a very well detailed way the problems that affect secure message transmission, as well as the attempts to address them. Throughout the manuscript, A. Di Falco et al. provide a valid and strong argument to corroborate the robustness of their protocols from both an analytical/numerical and experimental point of view. They further discuss all potential attacks that the protocol can be subjected to, always proving a security of almost 100%.

A. Di Falco et al. describe a protocol that is the classic version of the quantum key distribution (QKD) which is based on the BB84 protocol (invented by Bennet and Brassard in 1984 and that makes, in this work, use of photons). Indeed, A. Di Falco et al. show two communicating parts, i.e. Alice and Bob, as well as Eve, who try to intercept the message. As mentioned by the authors, a development of the quantum protocol BB84, despite guaranteeing much more security respect with their classical counterparts, is currently limited by, e.g.

the impossibility of amplifying photons, and so forth. Moreover, quantum-based communication protocols are significantly more expensive than their classical counterparts. With the results presented in the reviewed manuscript, the authors definitely pave the way towards the possibility of transmitting encrypted data and messages in a well-established and fiber-based classical communication network, while still preserving key security.

The reviewed work is novel and can have a significant impact not only on the scientific community, but also in terms of practical implementation for secure telecommunication over the existing infrastructure in a relatively immediate future. For these reasons, the manuscript “Perfect secrecy cryptography via correlated mixing of chaotic waves in irreversible time-varying silicon chips” by A. Di Falco et al. addresses the criteria for being published in Nature Communications.

AR: *We thank the Referee for the positive appreciation of the work and for the insightful comments provided in the report, which helped to improve the quality and outreach of the manuscript.*

RC: However, there are a few points that need to be addressed. As I mentioned before, the authors present the classic version of a QKD protocol only in the introduction, where they quickly describe the limits of quantum protocols. As a matter of fact, quantum protocols guarantee for more security than their classical counterparts due to the quantum mechanical laws (e.g. no cloning theorem, measurements that causes a collapse of the quantum states, thus allowing Alice and Bob to know whether their message has been intercepted or not). Furthermore, the ‘quantum revolution’ is making several steps towards the development of quantum-based technology. Therefore, at least in the description of the exchanged key between Alice and Bob, and in the evaluation of their ‘outcomes’, it is worth doing a parallel comparison with the quantum experiment. Of course, the system that the authors use cannot be adapted to the quantum regime. However, a parallel comparison between classical and quantum could make the manuscript also interesting and helpful for QKD protocols and thus for the quantum scientific community at large.

AR: *We followed the Referee’s suggestion and revised the manuscript by discussing this point in more detail. The summary of changes is as follows:*

- 1. We clarified in the revised text the connection between the original BB84 QKD scheme and the protocol derived in this work. As the Referee correctly pointed out, the classical scheme developed in the manuscript is inspired by the original BB84 QKD scheme based on the transmission of single photons states with random polarizations, here mimicked by random scattering events of a large ensemble of photons in the classical limit. The unique correlations between Alice and Bob in the quantum protocol are here achieved by the reciprocity theorem of electromagnetic waves, propagating in the classical bidirectional channel created by the two users. To clarify this connection, we added on lines 113–116:*

The communication protocol described in Fig. 1 can be regarded as a classical version of the original BB84 QKD scheme developed by Bennett and Brassard [45], in which the scatterers act as generator of random states, and the reciprocal communication line provides correlated measured states to the users Alice and Bob.

and added Ref. [45]:

- Bennett, C. H. & Brassard, G. Quantum cryptography: Public key distribution and coin tossing. In Proceedings of IEEE International Conference on Computers, Systems, and Signal Processing, 175 (1984)*

2. *We provided a quantum analysis of the classical protocol, as requested. This is indeed an interesting point. When a user, say Bob, launches a single photon in a random input position in his chip, the photon performs a random walk in the two scattering devices and emerges at a random position at Alice's end. If Alice injects the photon back in the same scattering channel, the reciprocity theorem of quantum mechanics guarantees that Bob measures the photon in the same channel that he originally used to sent the data.*

This process shares similarities with the quantum BB84 scheme, with scattering channels playing the role of random polarization states. However, there are also important differences. In order for the users to be able to use the scattering correlation to exchange bits, they are required to initially agree on a common dictionary that associates the same string of bit for correlated input-output positions in Alice and Bob's chips. This operation, not required in the classical limit (as the users measure the outcome of large ensemble of photons on all the channels), makes this limit potentially less interesting than the original BB84 scheme, which does not require this step as well. To clarify this point in the revised manuscript, we added on line 120–129:

In the quantum limit, when a user (say Bob), launches a single photon in the chip, the receiver (Alice) measures a photon emerging at a random position from the chip. If Alice injects the photon back in the same scattering channel, the reciprocity theorem of quantum mechanics [46] guarantees that Bob measures the emerging photon in the same input channel he originally used. This process shares some similarities to the quantum BB84 scheme, with scattering channels playing the role of random polarization states. However, there are also differences. For the users to exchange the same sequence of bits, they need to initially agree on a common dictionary that associates the same string of bit to correlated input-output positions in Alice and Bob's chips. This operation is not required in the classical limit (as the users measure the outcome of large ensemble of photons on all channels) and in the BB84 scheme.

and added Ref. [46]:

- *Deák, L. & Fülöp, T. Reciprocity in quantum, electromagnetic and other wave scattering. Annals of Physics 327, 1050 (2012).*

RC: The authors discuss the possibility of deterministic errors that might arise in the bitstream sequence between Alice and Bob due to active attacks. In this case, there is not information acquired by Eve during the attach resulting in a very secure cryptography system. Those errors are small and scale as the number of chaotic states available in Alice and Bob fingerprint devices. However, those need to be properly corrected in order to preserve the performance of the key distributions, the security, and the length of the cryptographic key. In this regard, the authors mention that techniques of information reconciliation and privacy amplification are available and can be integrated in their system, due to the public nature of the communication channel. The question that might arise, especially from those who are not familiar with error detection/correction techniques, is concerning whether or not these techniques are some sort of 'weakness' for the system, allowing potential security breach revealing the key. This point should be clarified by the authors.

AR: *We acted on the Referee's comment and revised the manuscript on lines 223–230:*

These errors can also be eliminated by using information reconciliation and privacy amplification [51-54], both conducted over the public authenticated channel. With information reconciliation, Alice and Bob obtain an identical key at each user's side by the exchange of minimal information (such as the mere bit parity of block key sequences). The second phase, privacy amplification, is then applied to eliminate effectively the information acquired by Eve during the reconciliation step. Privacy amplification is typically performed by using universal hash functions, which generate a new shorter key, on which Eve has zero information.

RC: General comments: The average Lyapunov exponent is labelled with the Greek letter 'Mu'. In the text there is an occurrence with 'Mu' being replaced by 'h'.

AR: *We revised the text and fixed the typo.*

RC: It is strongly suggested to check all the figures captions-legends for misprints.

AR: *We followed the Referee's suggestion and proofread carefully the manuscript, fixing all misprints.*

REVIEWERS' COMMENTS:

Reviewer #2 (Remarks to the Author):

The revised paper addressed all concerns that this reviewer had in the previous review. It is recommended that the paper be accepted for publication in its current form.

Reviewer #3 (Remarks to the Author):

In the manuscript titled "Perfect secrecy cryptography via correlated mixing of chaotic waves in irreversible time-varying silicon chips", the authors demonstrate the use of CMOS-compatible microring resonators to generate chaotic states of light in order to achieve secure cryptography. Specifically, the encryption key is exchanged via a one-time pad (OTP) technique, based on a classical Vernam cipher, thus via classical optical networks.

I acknowledge the changes that the authors implemented in the revised manuscript, which successfully address all my previous comments and suggestions in the original version. In particular, the authors added a very detailed and clear description of the difference between the quantum QKD scheme and its classical counterpart adopted in the manuscript. To further support this point, the authors have also included additional references. Moreover, they have also well explained the role of deterministic errors during communication and how this might be considered to preserve both information and security at the same time over the communication channel.

The authors have made substantial improvements in the content of the revised manuscript, which results in convincing outcomes that will significantly advance the knowledge in the field of cryptography for telecom applications.

A. Di Falco et al. have also addressed the comments and criticisms from Reviewers # 1 and 2. Therefore, I strongly recommend the manuscript titled "Perfect secrecy cryptography via correlated mixing of chaotic waves in irreversible time-varying silicon chips" by A. Di Falco et al. for publication in Nature Communications.

Author Response to Reviews of

Perfect secrecy cryptography via correlated mixing of chaotic waves in irreversible time-varying silicon chips

RC: Reviewer Comment, AR: Author Response, □ Manuscript text, “ ” Reference quotation

Reviewer #2 (Remarks to the Author)

RC: The revised paper addressed all concerns that this reviewer had in the previous review. It is recommended that the paper be accepted for publication in its current form.

AR: *We thank very much the Referee for the comment towards the publication of the paper.*

Reviewer #3 (Remarks to the Author):

RC: In the manuscript titled “Perfect secrecy cryptography via correlated mixing of chaotic waves in irreversible time-varying silicon chips”, the authors demonstrate the use of CMOS-compatible microring resonators to generate chaotic states of light in order to achieve secure cryptography. Specifically, the encryption key is exchanged via a one-time pad (OTP) technique, based on a classical Vernam cipher, thus via classical optical networks. I acknowledge the changes that the authors implemented in the revised manuscript, which successfully address all my previous comments and suggestions in the original version. In particular, the authors added a very detailed and clear description of the difference between the quantum QKD scheme and its classical counterpart adopted in the manuscript. To further support this point, the authors have also included additional references. Moreover, they have also well explained the role of deterministic errors during communication and how this might be considered to preserve both information and security at the same time over the communication channel.

The authors have made substantial improvements in the content of the revised manuscript, which results in convincing outcomes that will significantly advance the knowledge in the field of cryptography for telecom applications. A. Di Falco et al. have also addressed the comments and criticisms from Reviewers # 1 and 2. Therefore, I strongly recommend the manuscript titled "Perfect secrecy cryptography via correlated mixing of chaotic waves in irreversible time-varying silicon chips" by A. Di Falco et al. for publication in Nature Communications.

AR: *We thank very much Referee for the positive appreciation of our revision work and acceptance of the paper.*